# Maternal pertussis immunization and the blunting of routine vaccine effectiveness: a meta-analysis and modeling study

Michael Briga [1] ✉, Elizabeth Goult [1], Tobias S. Brett[2,3], Pejman Rohani[2,3,4] & Matthieu Domenech de Cellès [1]

A key goal of pertussis control is to protect infants too young to be vaccinated, the age group most vulnerable to this highly contagious respiratory infection. In the last decade, maternal immunization has been deployed in many countries, successfully reducing pertussis in this age group. Because of immunological blunting, however, this strategy may erode the effectiveness of primary vaccination at later ages. Here, we systematically reviewed the literature on the relative risk (RR) of pertussis after primary immunization of infants born to vaccinated vs. unvaccinated mothers. The four studies identified had ≤6 years of follow-up and large statistical uncertainty (meta-analysis weighted mean RR: 0.71, 95% CI: 0.38–1.32). To interpret this evidence, we designed a new mathematical model with explicit blunting mechanisms and evaluated maternal immunization's short- and long-term impact on pertussis transmission dynamics. We show that transient dynamics can mask blunting for at least a decade after rolling out maternal immunization. Hence, the current epidemiological evidence may be insufficient to rule out modest reductions in the effectiveness of primary vaccination. Irrespective of this potential collateral cost, we predict that maternal immunization will remain effective at protecting unvaccinated newborns, supporting current public health recommendations.

Pertussis is a highly transmissible respiratory infection that is primarily caused by the bacterium *Bordetella pertussis*[1,2]. Pertussis was a leading cause of childhood mortality until large-scale immunization programs, from the 1940s onwards, reduced pertussis notifications by over 90% in many countries[3,4]. However, in the last couple of decades, pertussis has re-emerged in many populations with high long-term immunization coverage, for reasons which remain highly debated[5].

Newborns are the age group most vulnerable to pertussis. In high-income countries, pertussis-related hospitalization rates in infants under 6 months are between 100 and 1000 per 100,000 per year[6], and data from the US[7] and Australia[8] show that over 60% of pertussis-associated hospitalizations are under one year old. To reduce the

burden of pertussis in vulnerable newborns, since 2012, many countries have introduced maternal immunization—i.e., vaccinating pregnant individuals, usually during the second or third trimester of pregnancy, with a low-dose tetanus toxoid, diphtheria toxoid, and acellular pertussis (Tdap) vaccine. In 2015, the World Health Organization (WHO) issued an official recommendation for maternal immunization with acellular vaccines against pertussis (based on studies with acellular primary immunization)[9], and by 2020 maternal immunization against pertussis was recommended in 55 countries[10].

Maternal immunization is highly effective at protecting newborns, with estimates of reductions in the risk of pertussis disease ranging from 70 to 95%[8,11–17]. However, the downstream consequences of

[1]Infectious Disease Epidemiology Group, Max Planck Institute for Infection Biology, Berlin, Germany. [2]Odum School of Ecology, University of Georgia, Athens, GA 30602, USA. [3]Department of Infectious Diseases, College of Veterinary Medicine, University of Georgia, Athens, GA 30602, USA. [4]Center of Ecology of Infectious Diseases, University of Georgia, Athens, GA 30602, USA. ✉e-mail: michbriga@gmail.com

maternal immunization, when infants receive their routine pertussis vaccines, are poorly understood. Specifically, there has been long-standing concern regarding potential immunological blunting, i.e., the interference of maternally transferred antibodies with the infant immune response[18–20]. Indeed, several studies and meta-analyses have shown that, after infants received their primary immunization, the antibody concentrations against several pertussis antigens were reduced by 30–60% in infants from vaccinated mothers relative to infants from unvaccinated mothers[21–24]. Similarly, the avidity of pertussis antibodies is reduced in infants from vaccinated mothers[25]. Interestingly, the blunting response following maternal pertussis immunization appears to be heterologous, also causing decreased antibody concentration after infants received a polio vaccine[26] and the blunting response also applies to other vaccines that contain (modified) diphtheria or tetanus toxins as carrier proteins, such as pneumococcal conjugate vaccines[22,26].

While immunological blunting is well documented, because the first maternal immunization programs were implemented in 2012, the long-term consequences of blunting on vaccine effectiveness (VE) and the ensuing epidemiology of pertussis remain difficult to evaluate. Here, we assess the epidemiological evidence for blunting and estimate the long-term consequences of maternal immunization in three steps. First, we perform a systematic review of the literature for estimates of the impact of maternal immunization on the relative risk of pertussis after primary pertussis immunization in infants from vaccinated mothers relative to unvaccinated mothers. Second, we extend a previously validated model of pertussis immunization[4,27] to simulate over several decades the short- and long-term epidemiological impact of maternal immunization with various levels of blunting and maternal immunization coverages on the age-specific time series of pertussis incidence. Third, to identify possible levels of blunting and the consequences of maternal immunization, we match the relative risk estimates obtained from the systematic review with those obtained from the modeling study.

## Results

### A systematic review of empirical studies on the relative risk of pertussis after maternal and infant primary immunization

We identified 374 peer-reviewed articles and, after removing 146 duplicate records, we screened 228 abstracts for a mention of maternal immunization with either a number of cases, an odds ratio or a relative risk of pertussis and/or effectiveness of a pertussis vaccine (Fig. 1). Based on these criteria, we identified and retrieved 69 articles that might contain information (Table S4).

Our search identified four studies that reported five estimates of the relative risk of pertussis in infants from vaccinated versus unvaccinated mothers and that specified the dose of infant primary immunization (Figs. 1 and 2, and Table S1). Two studies were carried out in the UK, one in Australia, and one in California, US. In all studies, maternal immunization coverage was low right after the implementation of maternal immunization but increased to reach >70% (Table S1). The studies' follow-up times ranged from 2 to 6 years (Table S1). At the first dose of infant primary immunization, these studies showed a weighted mean RR of 0.26 (Fig. 2, 95% CI: 0.11–0.67). At the second dose of infant primary immunization, the RR increased to a weighted mean of 0.73 (Fig. 2, 95% CI: 0.39–1.34) and this was very similar to the weighted mean RR at the third dose of infant primary immunization, which had a value of 0.71 (Fig. 2, 95% CI: 0.38–1.32).

### The model recapitulates the historical trends of pertussis, with a resurgence caused by a honeymoon effect

To interpret the RR estimates and assess the short- and long-term impact of blunting on pertussis transmission dynamics, we extended an empirically validated model of pertussis epidemiology[4,27], to incorporate maternal immunization and its dual effects of protecting unvaccinated newborns and blunting vaccine effectiveness after infant primary immunization. To assess the historical impact of routine immunization in infants, we first examined the trends in overall incidence and age-specific susceptibility. Infant immunization induced a

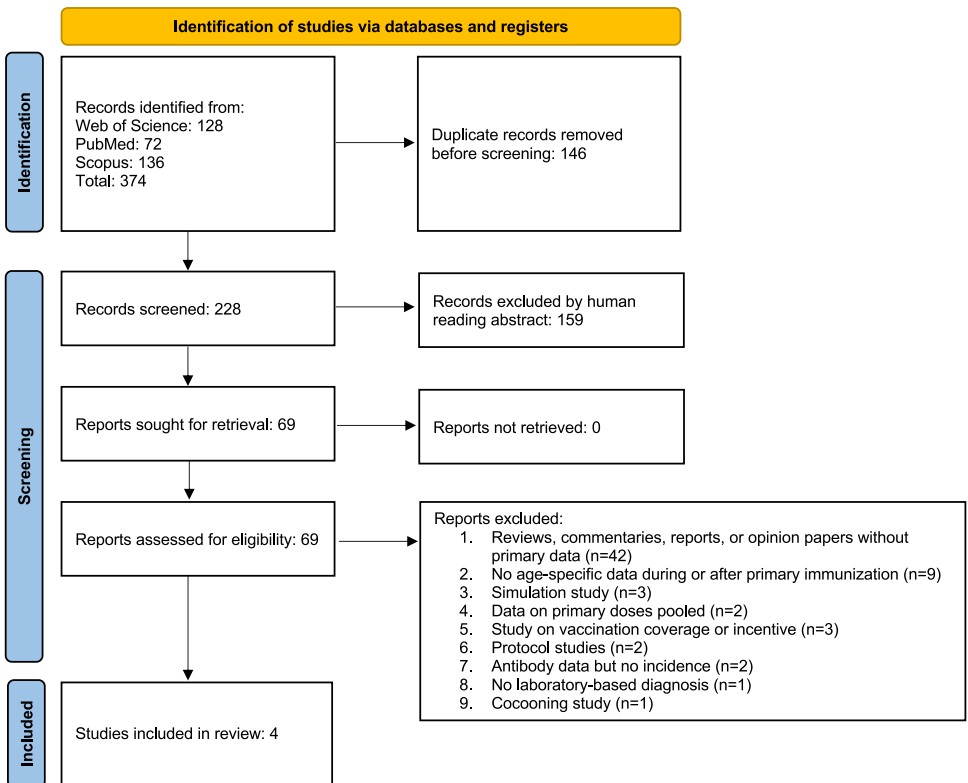

**Fig. 1** | Prisma flow chart, following the PRISMA guidelines[72], of the literature search for empirical estimates of the relative risk (RR) of contracting pertussis following maternal immunization in children who had received at least one dose of infant primary immunization.

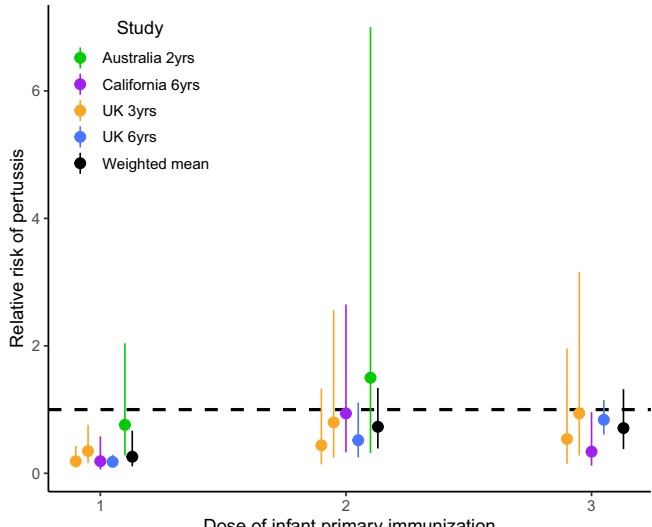

**Fig. 2 | Overview of the results from empirical epidemiologic studies of the relative risk of contracting pertussis after 1–3 vaccine doses for primary immunization in infants from vaccinated mothers relative to those of unvaccinated mothers during infant primary immunization.** The literature search protocol is shown in Fig. 1, and the data and sample sizes for each estimate are available in Table S1. The two lines for the UK 3-year study represent two estimates of maternal immunization coverage. Dots represent means and error bars show 95% CI. References: Australia 2 yrs[16], California 6 yrs[14], UK 3 yrs[13], and UK 6 yrs[17]. yrs = years.

strong decrease in pertussis incidence but was followed by a rebound with a gradual buildup of susceptible individuals in adult age groups until a new equilibrium—with lower incidence than in the pre-vaccine era—was reached several decades later (Fig. 3A, B). As previously demonstrated in the US, this resurgence was explained as an "end-of-honeymoon" effect, a predictable consequence of incomplete immunization with imperfect but highly effective vaccines. Hence, our model recapitulated the historical dynamics of pertussis in the US, characterized by a resurgence and shift of infections to adolescent and adult age groups[4].

**Transient dynamics after the start of maternal immunization**
Second, we examined the impact of maternal immunization, which we introduced 100 years after the rollout of infant immunization, to disentangle the consequences of the two immunization programs. Maternal immunization had the intended effect of decreasing susceptibility to infection in unvaccinated newborns aged 0–2 months (Fig. 3B). However, after the rollout of maternal immunization, the model predicted a transient phase with lower pertussis incidence, lasting at least 5 years and followed by a rebound (Fig. 4A). In newborns, pertussis incidence was lower after the start of maternal immunization relative to before, but the benefit of maternal immunization was larger in a scenario without blunting than in a scenario with blunting (Fig. 4A). Thus, maternal immunization was predicted to be effective at protecting unvaccinated newborns, with benefits by far outweighing any possible blunting-mediated increases in incidence.

By contrast, in the second age class of infants aged 3–18 months, (i.e., after receipt of the primary pertussis immunization), maternal immunization was followed by a decrease of pertussis incidence, but only when blunting was low (<10%, Fig. 4B). At higher levels of blunting, pertussis incidence was predicted to increase relative to before the start of maternal immunization (Fig. 4B). Irrespective of the simulated blunting level, the second age class also showed a clear transient phase, lasting at least 5 years (Fig. 4B). During this transient phase, pertussis incidence was first predominantly driven by infants from unvaccinated mothers (Fig. 4C), followed later on by incidence in

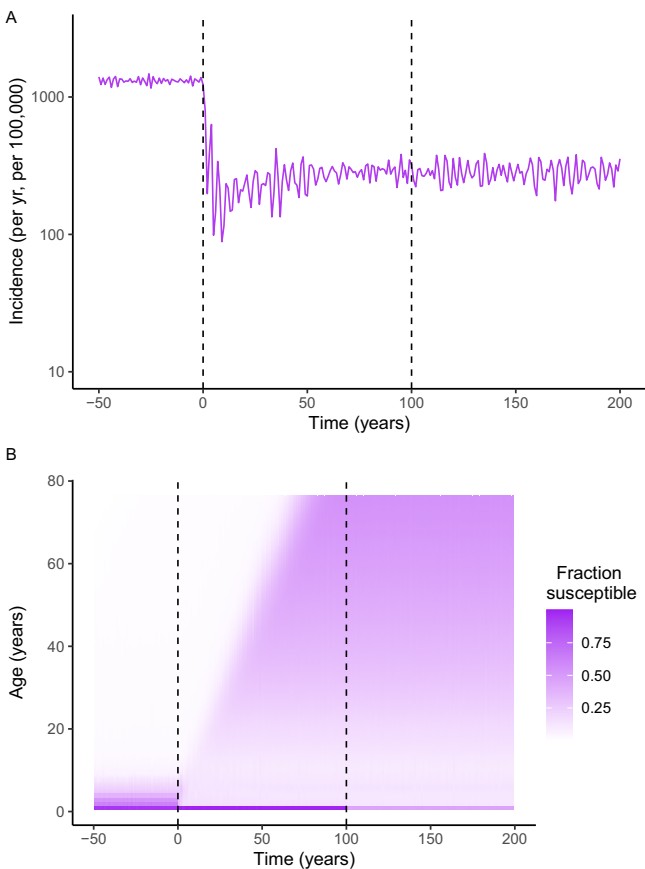

**Fig. 3 | Illustration of honeymoon effects and transient susceptible dynamics following the start of infant and maternal immunization programs, based one randomly selected stochastic simulation.** At the start of infant immunization (vertical dashed line at time = 0 years) and maternal immunization without blunting (vertical dashed line at time = 100 years) the simulation shows transient dynamics for (**A**) pertussis incidence and (**B**) the fraction susceptible for each age class from one stochastic simulation.

infants from vaccinated mothers, who dominated once incidence had reached equilibrium (Fig. 4D). These dynamics persisted in children up to 5 years of age, but dissipated by age 10 years, after which pertussis incidence remained low and the effects of both maternal immunization and blunting on pertussis incidence were negligible (Fig. S3).

**Comparison with empirical estimates suggests that blunting cannot be ruled out**
Lastly, we estimated the impact of blunting on the RR of pertussis in vaccinated infants born to mothers vaccinated during pregnancy. In newborns, our simulations show that estimates of the effectiveness of maternal immunization (i.e., based on incidence) have a transient phase in a direction consistent with those deduced from the transient incidence dynamics: the effectiveness of maternal immunization was high at first but decreased with time, and equilibrium values were reached only after a transient phase that lasted at least a decade (Fig. 5A). Hence, ignoring the transient dynamics results in an overestimation of the effectiveness of maternal immunization, an effect most pronounced during the first decade after implementation.

As expected, in the second age class of infants aged 3–18 months (i.e., after receipt of the primary pertussis immunization), blunting resulted in RR equilibrium values above 1 (i.e., a higher risk of pertussis in infants born to vaccinated mothers relative to unvaccinated mothers), which increased with the input blunting strength. However, these risks were reached only after a transient phase that lasted more than a decade (Fig. 5B). During this transient phase, even in scenarios

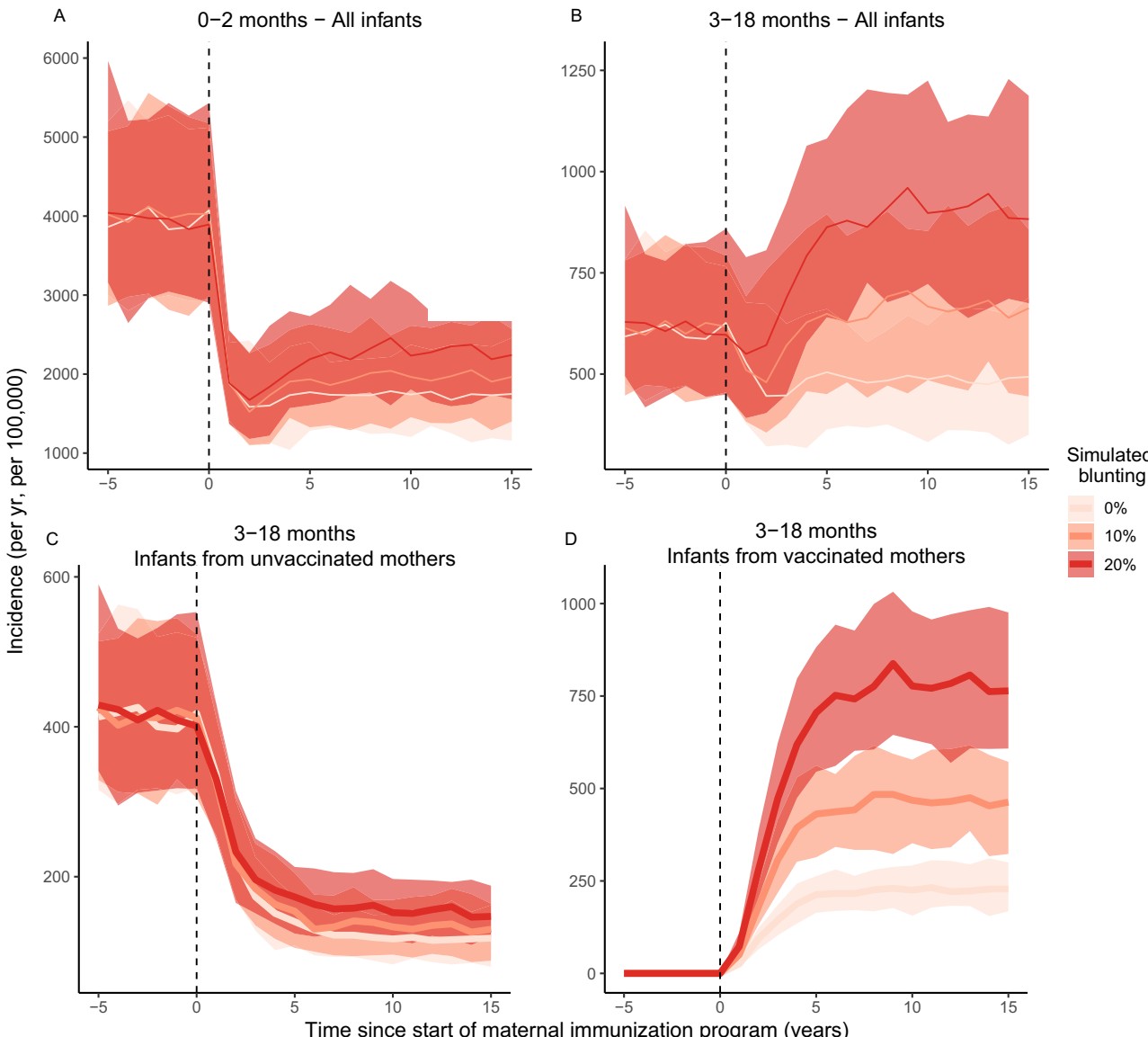

**Fig. 4 | Model predictions of the consequences of maternal immunization and blunting for pertussis incidence.** Panels show infants of different ages and vaccination history, with (**A**) unvaccinated newborns (aged 0–2 mo) and (**B**) vaccinated infants (aged 3–18 mo, after receipt of primary immunization). For this age class, during the transient phase after maternal immunization, pertussis incidence is at first mostly driven by (**C**) infants from unvaccinated mothers until (**D**) pertussis incidence is most abundant in infants from vaccinated mothers. Panel (**D**) is consistent with the PCV-term in the RR equation (2) (see "Methods" section[67]). Solid lines show the median incidence of 100 simulations and shaded areas represent the 95% CI. Pertussis incidence before the start of the maternal immunization program is shown as a control, and variation in incidence between simulations is the result of demographic stochasticity. In this figure, all immunization coverages were set at 70%, which is close to the mean observed values in empirical studies (e.g., Table S1). Note the different y-axes between panels.

with blunting, the RR was initially far below 1 and gradually increased after the start of maternal immunization. Hence, ignoring these transient dynamics and assuming that an RR of 1 indicates no blunting may result in overestimating the effectiveness of maternal immunization in both unvaccinated newborns and vaccinated infants (Fig. 5B). It is noteworthy that the transient phase for the estimation of RR (Fig. 5A, B) is longer than for the incidence (Fig. 4A, B). This is because RR (and hence VE) are cumulative estimates instead of instantaneous estimates such as incidence (Fig. S4). Interestingly, depending on the contact matrix, the incidence during the transient phase showed oscillating dynamics, which increases the variability between estimates after 3, 6, or even 10 years (Figs. S12 and S13), but these oscillations disappeared in the RR estimates (Figs. S14 and S15), likely as a result of their cumulative nature. Interestingly, after the transient phase, a scenario without blunting resulted in RRs below 1, due to residual protection

from maternal antibodies in infants who were vaccinated but in whom the vaccine did not take. This result suggests that the usual assumption that an RR of 1 indicates no blunting may be too optimistic.

Comparing our model predictions with the results of epidemiological studies, all current empirical RR estimates fell well within the transient phase of our simulations (Fig. 5C). Demographic stochasticity accounted for some of the variability in the predicted RR (Fig. 5B), but observation noise due to small sample sizes was by far the largest source of uncertainty (Fig. 5C). Inspecting the overlap between confidence intervals, all the empirical estimates were consistent with blunting, even though two estimates (in the UK and California) were also consistent with no blunting. Hence, according to our simulations, current empirical studies cannot yet detect the eventual epidemiological consequences of blunting (Fig. 5B), because of the transient RR dynamics following the rollout of maternal immunization. Hence, the model-data comparison

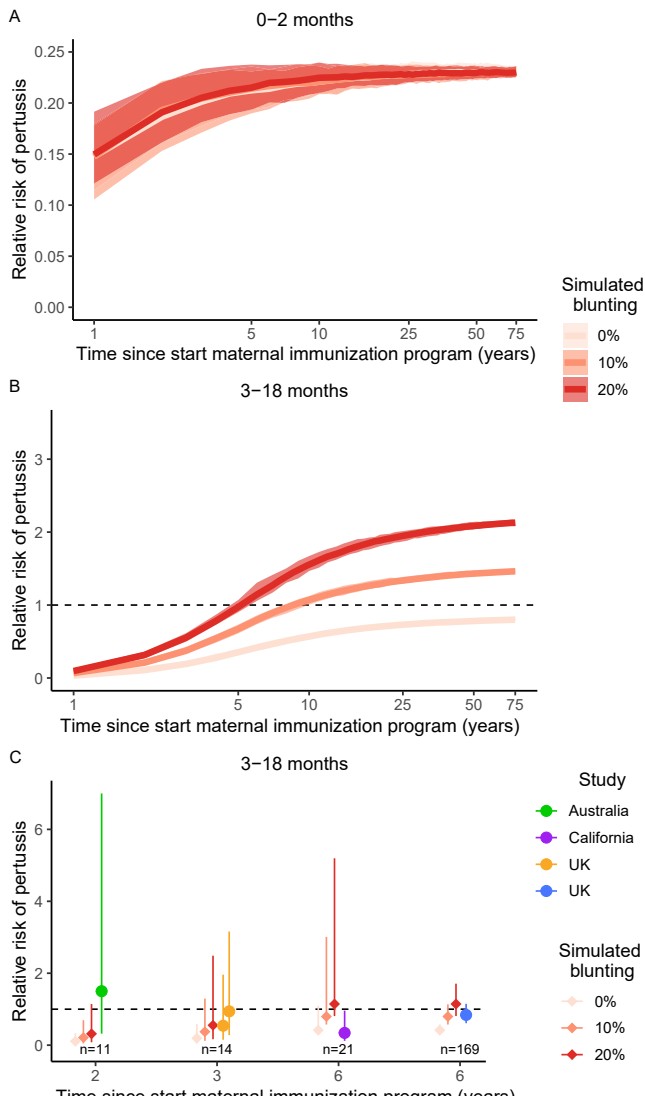

**Fig. 5 | Temporal dynamics of the risk of contracting pertussis following the start of the maternal immunization program.** Panels show infants of different ages and vaccination history, with (**A**) unvaccinated newborns and (**B**, **C**) vaccinated infants from vaccinated mothers relative to those of unvaccinated mothers. **A**, **B** Solid lines show the median RR of 100 simulations and shaded areas show 95% CI resulting from demographic stochasticity across simulations. **C** Comparison between the results from model simulations (gradient red diamonds) and the empirical studies (colored dots). In panel **C**, dots represent means and vertical lines show 95% CI for both data and simulations. Here, the 95% CI of the simulations emerge from both demographic stochasticity and observation uncertainty, resulting from the partial sampling of the population with the sample sizes (number of cases) matching those from empirical studies in Fig. 2 (Table S1). In panels A–B, the x-axis is log transformed to highlight the time period shortly after the start of maternal immunization.

suggests that current empirical RR estimates cannot rule out a modest level of blunting of infant primary immunization (Fig. 5C).

**Sensitivity analyses**

To assess the robustness of our results, we conducted a range of sensitivity analyses with different hypotheses regarding model parameters, such as the coverage of maternal and infant immunization, the average duration of maternal protection, age-specific contact patterns, the age of primary immunization, and the basic reproduction number. To assess the consequences of non-pharmaceutical interventions

(NPIs) during the COVID-19 pandemic, we further simulated scenarios with transient reductions in vaccine coverage and pertussis transmission. In our simulations, variation in these parameters sometimes had a substantial impact on pertussis incidence, e.g., Figs. S7, S9, S11, and S13. However, our conclusions about blunting were robust in all the sensitivity analyses, and the results presented in Figs. S5–S34 were consistent with those presented in Figs. 4 and 5.

The sensitivity analyses revealed a few noteworthy results. First, a considerable reduction of the blunting effect on pertussis incidence was predicted when delaying the age at the start of the infant's primary immunization by a few months (Figs. S11 and S12). Second, the empirical epidemiological studies with two or three years of monitoring could match scenarios with blunting reducing vaccine effectiveness by up to 50% (Fig. S22). However, such a high level of blunting could be ruled out by the 6-year study in the UK. This indicates the need for larger sample sizes and longer monitoring to reliably estimate vaccine effectiveness and the impact of new immunization programs. Third, consistent with expectation and epidemiological observations, the COVID-19 pandemic was predicted to have a substantial impact on pertussis dynamics, with a decline in incidence followed by a rebound (e.g., Figs. S25A, B, S27A, B, S29A, B, S31A, B, and S34A–D). However, the temporal dynamics of RR remained consistent in this scenario (Figs. S32, S33, and S34E–H), because in our model control measures equally affected infants born from vaccinated or unvaccinated mothers.

## Discussion

In this study, we aimed to review the epidemiological evidence regarding the impact of maternal immunization on the effectiveness of pertussis vaccines in infants. We interpreted this evidence using a new mechanistic mathematical model that represented the effect of blunting on pertussis transmission dynamics. Our systematic review identified four observational epidemiological studies, which had a maximum monitoring time of 6 years and small sample sizes. Most studies suggested a reduced risk of pertussis after receipt of the first vaccine dose in infants born to mothers immunized during pregnancy. After receipt of the second or third vaccine dose, however, the estimates had large uncertainty, consistent with a range of assumptions about blunting levels. To interpret these empirical estimates, we performed a simulation study using an age-structured model of pertussis transmission. This model predicted that the introduction of maternal immunization was followed by a transient phase lasting at least a decade, during which the blunting effect of maternal antibodies could be underestimated. The model-data comparison suggested that a modest level of blunting could not be ruled out, even though the large uncertainty in empirical estimates prevented a definitive conclusion. Even in the presence of such blunting, however, our model predicted that maternal immunization remained effective at reducing pertussis in unvaccinated newborns. Hence, our results confirm that maternal immunization is an effective strategy to protect this age group, but suggest it may eventually result in an infection-control trade-off with older age groups.

Unvaccinated newborns are the age group most vulnerable to pertussis, both in terms of infection and hospitalization risks[6–8]. Our study suggests that, although blunting may erode the benefits of immunization in infants, maternal immunization is highly effective at protecting unvaccinated newborns. Hence, our results support the decision of many public health authorities to introduce maternal immunization against pertussis, provided that the main control objective is to protect newborns[9,10]. More generally, given the vulnerability of newborns to many pathogens, maternal immunization against other infectious diseases is considered by many public health authorities, and the pharmaceutical industry is developing new vaccines for immunization during pregnancy. For example, a maternal vaccine against RSV recently underwent a successful phase 3 trial[28] and may soon be rolled out[29]. In addition to the Tdap and RSV vaccines, maternal immunization is implemented or recommended for influenza, COVID-19, and several

other infections[30–32]. Maternal immunization, therefore, also appears as a promising strategy to protect newborns against these infections, but given that immunological blunting has been documented for many infections[18,33–36], our results suggest the need to monitor its impact on subsequent vaccination of infants carefully.

One of the main takeaways from our review was the large uncertainty around the available empirical estimates, emphasizing the urgent need for more research on blunting. However, if such blunting is confirmed by future evidence, there are ways to mitigate its impact. Immunity in newborns conferred by maternal antibodies against pertussis likely wanes quickly, on a timescale of months, according to our model. Given the fast waning of maternal antibodies and the association between the maternal antibody titers and the blunting of infant immune responses[37], delaying infant primary immunization by a few months might greatly reduce the maternal blunting of infant immunization[35]. Pertussis immunization schedules vary greatly between countries, with the age at first immunization of infants typically ranging between two and four months[38], and the current variation in pertussis immunization schedule can significantly impact the effectiveness of pertussis immunization in infants[39]. If maternal blunting occurs, our results suggest that the optimal age at first immunization is governed by a trade-off between vaccinating too early (with the effect of reducing the susceptibility age window in newborns but amplifying the impact of blunting in vaccinated infants) and too late (with the opposite effect). As the costs of infection in unvaccinated newborns likely exceed those in older age groups, the introduction of maternal immunization may thus increase the optimal age at first immunization, in all infants or only in those born to vaccinated mothers. The Dutch public health authorities take an interesting approach: the recommended age for the first dose of primary immunization for children from unvaccinated mothers is two months, while for children from vaccinated mothers, the recommendation is to start one month later, at three months[40]. Identifying the optimal pertussis immunization schedule for infants would be helpful for public health authorities, especially given the vast between-country variation in pertussis immunization schedules[38,39,41]. Our new model could thus help future research on optimal vaccine schedules.

After the roll-out of maternal immunization, our model predicted a transient phase lasting approximately a decade, during which the incidence of pertussis—in both unvaccinated newborns and vaccinated infants—initially dropped, but then bounced back and increased to reach a new equilibrium. The duration of this transient phase is consistent with that found in a previous modeling study[42] and may be explained by the time required for the first cohorts of blunted-vaccinated infants (i.e., born to vaccinated mothers) to reach primary school age and its associated high contacts. Thus, reduced transmission from unvaccinated newborns may explain the initial decrease in incidence across age groups, but this benefit gradually wears off as blunted-vaccinated, and therefore less well-protected, children age.

From a practical perspective, the transient phase following maternal immunization has at least two important implications. First, the current individual-based epidemiological studies had ≤6 years of monitoring, which we predict is insufficient to capture the consequences of maternal immunization that can only be detected with at least a decade of data. Second, an RR of 1 is considered the baseline value indicating that maternal immunization does not blunt vaccine effectiveness in infants. In the transient phase, however, our model predicts that the baseline value under a no-blunting scenario starts off close to 0 and gradually increases to an equilibrium just below 1. This equilibrium value results from the residual protection from maternal antibodies, which partially compensates for failures of primary immunization in infants born to vaccinated mothers. These results suggest that the early empirical estimates of vaccine effectiveness may overestimate the benefits of maternal immunization, both for newborns and for infants. A testable prediction of our model, therefore, is that the benefits of maternal immunization estimated in subsequent studies will be lower than those currently reported, both in newborns and after the primary series. More broadly, as reflected in some official guidelines for vaccine impact evaluation[43], biased estimation of vaccine effectiveness because of transient dynamics early after vaccine rollout is likely a more general occurrence.

Epidemiological monitoring studies have, until now, not reported maternal immunization-mediated rebounds in pertussis incidence (e.g.[44]). However, we believe that the best way to quantify the clinical and epidemiological impact of blunting is by more precise individual-level estimation of the RR (ideally after the transient phase predicted by our model), instead of monitoring population-level changes in pertussis incidence. This is for several reasons. First, we predict that the rebound a few years after the roll-out of maternal immunization will be small compared to the large reduction immediately after. Hence, detecting an increase in pertussis incidence resulting from maternal immunization may be difficult. Second, many countries have reported pertussis resurgence in the last decades[5], so empirical studies of pertussis incidence may need to disentangle pre-maternal immunization increases from a post-maternal immunization rebound, which is challenging, especially considering the multiannual periodicity of pertussis. The interpretation of trends in pertussis incidence may be further compounded by changes in national immunization schedules (e.g.[39]), and the impact of NPIs during the COVID-19 pandemic. Hence, we predict that the RR will remain a more robust metric.

In addition to the monitoring time, other components of the design of current epidemiological studies may affect their interpretation. First, as the mothers' decision to receive Tdap during pregnancy is voluntary and likely related to other socioeconomic factors that will affect—and presumably reduce—their infants' risk of pertussis, the baseline comparability between groups in these observational studies is far from obvious. Hence, estimates of the effectiveness of maternal immunization may have been confounded by healthy user bias; interestingly, the only study that controlled for maternal characteristics[16] had large statistical uncertainty around its RR estimates and could not rule out blunting. In addition to this confounding problem, maternal immunization may prevent mothers from transmitting pertussis to their infants, an indirect effect that could lead to underestimating the risk of pertussis and masking the effect of blunting in the group of infants vaccinated and born to vaccinated mothers (as recognized by some investigators[13]). Because of these potential biases, current RR estimates may have been underestimated. Our model does not account for such potential biases, and hence, our model predictions regarding the level of blunting may be conservative.

Our study has several limitations. First, our model was parameterized based on previous estimates in Massachusetts, USA, and may only correctly represent pertussis epidemiology in comparable high-income countries. To address this limitation, we have performed sensitivity analyses with a range of basic reproduction numbers, and the conclusions were consistent. Second, in the absence of serological correlates of protection for pertussis[45–47], the duration of maternally-derived protection—and in particular, how it connects to maternal antibody titers in the infant—is unknown. In our model, this parameter was calibrated to reach empirical estimates of maternal immunization effectiveness in newborns, with an average duration of maternally derived protection at 8.7 months resulting in vaccine effectiveness for newborns close to 80%. Even though sensitivity analyses demonstrated that our main results were robust to variations in this parameter, more accurate estimation will be important to inform future models and to predict optimal vaccination schedules. Third, for simplicity, we did not model the direct, protective effect of maternal immunization on mothers—only the indirect effect on their newborns. This direct effect is an additional benefit of maternal immunization, but because in high-income countries the proportion of women giving birth is small relative to the total adult population (the 2010–2021 EU average ranging between 1.50 and 1.57 live births per 1000 individuals

and the range among all EU-member states in 2021 between 1.1 and 1.8 live births per 1000 individuals[48]), it is expected to have a minor epidemiological impact. Fourth, in our simulations, maternal immunization coverage was fixed at 70%, while in real-world settings it is often initially low and takes several years to increase to the higher values described in Table S1 (e.g., Fig. 1 in ref. 14). However, such temporal variations in immunization coverage are expected to increase the duration of the transient phase and would thus reinforce our conclusions about the unreliability of early empirical estimates.

To conclude, our study shows that maternal immunization is effective at protecting newborns, even when blunting erodes some of its benefits. The degree of blunting and the extent to which it entails an epidemiological cost in older age groups can not be known yet, as we predict that the implementation of maternal immunization is followed by a transient phase during which the epidemiological impact of blunting is masked. Hence, current epidemiological studies may be insufficient to rule out blunting or grasp the long-term epidemiological consequences of maternal immunization. Our results, therefore, identify the need for more research to precisely estimate the degree of blunting after primary immunization, if any. More generally, our study supports the use of maternal immunization to reduce pertussis in newborns, but suggests this strategy may be associated with an infection-control trade-off between different age groups.

## Methods

### Systematic review and meta-analysis of empirical studies on the relative risk of pertussis after maternal and infant primary immunization

We searched the literature on Web of Science, PubMed, and Scopus on August 25, 2023. We used the search terms "maternal immunization AND pertussis AND effectiveness" OR "maternal vaccination AND pertussis AND effectiveness". In Web of Science, we entered these terms under 'Topic', in PubMed we entered these terms under 'Title/Abstract', and in Scopus under 'Article type, Abstract, Keywords'. In these three searches, we selected the publication dates from 1 January 2012, which is the first complete year with recommended maternal immunization in any country[14], until 25 August 2023. To be included in our review, studies had to provide an estimate of the relative risk (RR) of pertussis in infants having received at least one dose of their primary immunization from vaccinated vs. unvaccinated mothers. We selected only those studies that used laboratory-confirmed diagnosis of pertussis. Because the RR of pertussis varies substantially between doses of infant primary immunization, we selected only studies that were explicit about the number of doses of primary immunization and we excluded those studies that pooled estimates for various doses. For studies that provided more than one estimate per dose, we used all estimates provided. When studies provided estimates with and without known pertussis onset dates, we chose those estimates with known onset dates.

To estimate the RR per dose of infant primary immunization, we performed the analyses in two ways, which gave consistent results. First, we estimated the weighted means of RR per dose. Second, we performed a meta-regression following[49] and as performed in[50]. In brief, in both analyses, we log-transformed the RR obtained from the studies and weighted the estimates according to the inverse of the 95% CI of the RR estimate. We preferred to use RR rather than the number of cases[51] because there are two studies in which RR estimates correct for participants' socio-demographic covariates. The meta-regression allowed the inclusion of all doses (as a factor) into one model and to account for estimates coming from the same studies by having the study identity as a random intercept. To account for the fact that two studies were carried out in the same population, we repeated the meta-regression by including the population as a random intercept, which gave consistent results. We performed the meta-regression in R v. 4.2.0[52] with the function 'brm' from the package 'brms'[53] using flat uninformative priors with 100,000 iterations, 20,000 burn-ins, and a

thinning of 40. Model residuals fulfilled all assumptions as checked with the function 'createDHARMA' of the package 'DHARMa'[54]. Meta-analyses usually control for the heterogeneity between studies and, most importantly, for publication bias[49], but given the low number of studies and sample sizes, this was not possible here.

### Model description

We implemented an age-structured model of pertussis transmission, extending a previous model empirically validated on data from Massachusetts, USA[4,27]. In brief, the model is based on the Susceptible-Exposed-Infected-Recovery (SEIR, Fig. S1) model in which pertussis vaccines can fail by (i) failure in "take" (primary vaccine failure) and (ii) failure in duration (waning of vaccine protection)[55]. Based on immunological evidence showing an immediate reduction in the antibody response to primary vaccination[21,22,35], we assumed that blunting from maternal immunization increased the probability of primary vaccine failure. Previous results in[4] found no evidence for failure in degree (or leakiness, i.e., when vaccine-induced protection is imperfect and vaccinees remain susceptible to infection, but at a lower degree than unvaccinated individuals), and hence we ignored this possibility.

The model is designed such that we follow individuals grouped according to their immunization history. For each immunization, individuals can go to either of three compartments: one for successful immunization, one for failed immunization, and one for immunization not received (Fig. S1). These three possible paths or compartments start from their mother's immunization status during pregnancy, followed by an infant immunization schedule that resembles that of the empirical studies in Table S1: the infant's primary immunization occurs at the age of three months, and an infant booster at the age of 1.5 years. Hence, newborns can be born in three possible compartments: from vaccinated mothers whose immunization succeeded, mothers whose immunization failed (i.e., who received the vaccine but whose infant remained unprotected), or unvaccinated mothers. We model maternal immunization by transferring a fraction of newborns to a protected class. Maternal immunization in itself is not modeled; hence our model does not specify at which stage of pregnancy maternal immunization occurs. The assumed high effectiveness of maternal immunization implies that it occurred at a timing consistent with that used in the studies in Table S1 and in the empirical studies of the systematic review, i.e., during the second or third trimester of pregnancy and at least one week before giving birth.

Each of the three compartments is followed by a compartment for successful primary infant immunization (Fig. S1) and a compartment for failed primary infant immunization, thereby becoming susceptible (Fig. S1), or no immunization thereby also becoming susceptible. Following the first booster, these susceptibles become immunized again at an effectiveness of 96%, with a subsequent waning rate of vaccine protection of 0.011 yr$^{-1}$ (Table S3). For simplicity, as in earlier modeling studies[4], we model the infant primary immunization as one event and we do not consider the gradual effect of multiple vaccine doses.

### Model parametrization

**Parametrization of the base model.** Our model is based on ref. 4, which was fitted to pertussis incidence data from Massachusetts to identify the model parameters that best matched the data. Hence, for the parameters consistent between both models, we used the estimates from ref. 4 (Table S3), but note that the sensitivity analyses explained below and shown in the supplementary information create conditions that represent other populations.

By adding maternal immunization to this model, we also added three new parameters. The first parameter is maternal immunization coverage, for which we use a baseline of 70%, consistent with the empirical estimates a few years after the rollout of maternal immunization (Table S1). The second parameter is the duration of maternally derived immunity, for which there are no empirical estimates. One

proxy is the half-life of maternal antibodies, which ranges between 29 and 36 days[56,57], suggesting that the duration of maternally-derived immunity is short. However, because of the absence of definite correlates of protection[45-47], the correspondence between the waning rate of maternal antibodies and that of maternal immunity is unknown. For example, if both waning rates are assumed equal and without the transfer of new maternal antibodies, e.g., through breastfeeding, according to our model the effectiveness of maternal immunization in the first two months after birth would be close to 50% (Fig. S1). This is in contrast with the estimated effectiveness of maternal immunization in that age class, which ranged from 78% ([15] 95% CI: 48–90) to 93% ([12] 95% CI: 81–97). Thus, in our model, we calibrated the average duration of maternal protection to reach these empirical estimates of vaccine effectiveness and hence we show results with an average duration of 8.7 months (Table S3 and Fig. S2). The third parameter is the blunting of infant immune protection, defined as the relative reduction in the effectiveness of the primary series in infants born to vaccinated mothers compared to infants born to unvaccinated mothers. There are no empirical estimates of the degree of blunting of infant protection following maternal immunization or how changes in IgG titers translate into changes in protection. When infants receive their primary immunization, the antibody concentrations of infants from vaccinated mothers are 30% to 60% lower compared to those from unvaccinated mothers[21,22]. To best match the empirical estimates of the relative risk of pertussis, we chose a decrease in the effectiveness of primary immunization from 0% (i.e., no blunting) to 20%. Formally, the blunting parameter $b_1$ is modeled as a relative reduction in the initial effectiveness of infant immunization $\varepsilon$, resulting in a blunted effectiveness

$$\bar{\varepsilon} = \varepsilon(1 - b_1) \qquad (1)$$

**Sensitivity analyses.** We checked the sensitivity of our results to nine model changes:
(i) maternal immunization coverage, changed from a baseline value of 70% to 50% and to 90% (Figs. S5 and S6), which is the range represented in Table S1;
(ii) infant immunization coverage, changed from a baseline value of 90% to 70% and 80% (Figs. S7 and S8);
(iii) the average duration of maternal protection changed from a baseline value of 8.7 months to 4 months and 1 year (Figs. S9 and S10);
(iv) the start of primary immunization, changed from a baseline value of 3 months to 2, 4, and 9 months (Figs. S11 and S12);
(v) different social contact matrices, with that of the UK as baseline (in Figs. 4 and 5) and changed to those in 7 other European countries as described in[58,59] (Figs. S13–S18). A more detailed age-specific contact matrix is available for the UK[60], but not for other countries, and hence we used an age structure that was consistent across countries;
(vi) starting the maternal immunization program 60 years after the rollout of primary immunization, instead of 100 years. This reflects an introduction of maternal immunization during a honeymoon-mediated pertussis resurgence, as was the case in many countries (Figs. S19 and S20);
(vii) blunting the effectiveness of the infant primary immunization up to 60% (Figs. S21 and S22), instead of the baseline range of 0–20%;
(viii) ±50% variations in the basic reproduction number, to model different transmission levels in a range of populations (Figs. S23 and S24);
(ix) the impact of the COVID-19 pandemic, modeled as a 3- to 9-month period associated with lower vaccine coverage (20% reduction) and lower transmission (20%, 50%, or 70% reduction, as a result of social distancing measures, Figs. S25–S34). This sensitivity analysis is motivated by the observation that COVID-19-associated social distancing measures substantially decreased

pertussis incidence[61,62]. To model NPIs during the COVID-19 pandemic, we followed the approach from a previous study which reduced the transmission between 20% and 70% for a duration of 3 months and 9 months[63]. We obtained estimates of the changes in immunization coverage during the COVID-19 pandemic from three empirical studies[64-66]. In our model, we introduced these changes 8 years after the rollout of the maternal immunization program because that matches the 8-year lag between the first maternal immunization programs and the COVID-19 NPIs in most countries.

### Estimation of relative risk of infection following maternal immunization
To connect the model outputs with the empirical estimates of relative risk (RR)[13,14,16,17], we calculated the risk of contracting pertussis in infants born to vaccinated mothers relative to that of infants born to unvaccinated mothers using the same approach as in[13,17]. These studies used the screening method following[67], in which the RR is estimated as:

$$RR = \frac{PCV}{(1 - PCV)} \cdot \frac{(1 - PPV)}{PPV} \qquad (2)$$

where PCV (proportion of cases vaccinated) is the proportion of cases in vaccinated infants born to vaccinated mothers among all cases in infants vaccinated, and PPV is the proportion of the population vaccinated, here equal to the vaccine coverage of maternal immunization. We validated the RR estimates of the simulations by setting the half-life of maternally derived protection to infinity, which resulted in an RR of 0.05 in newborns and matched the initial 95% effectiveness of maternal immunization against pertussis in unvaccinated newborns fixed in the simulations (Fig. S2 and Table S3). Hence, while the screening method may have several limitations in real-world settings[68], it provided reliable estimates of vaccine effectiveness in the context of our model.

### Numerical implementation
We represented the process model as a continuous-time Markov process, approximated via the tau-leap algorithm[69] with a fixed time step of 1 day. To compare the model outputs to empirical estimates, we also added an observation model for the RR. This was done by sampling the number of vaccinated cases born to vaccinated mothers from a Binomial distribution with probability PCV and size equal to the number of cases described in the empirical studies at the third dose (Table S1). The model was implemented and simulated using the 'pomp' package v. 4.4[70], operating in R v. 4.2.0[52].

### Reporting summary
Further information on research design is available in the Nature Portfolio Reporting Summary linked to this article.

## Data availability
The data extracted in the systematic review is included in Supplementary Table S1 and all other data analysed are available in the R scripts.

## Code availability
All R scripts are freely available via the Harvard dataverse[71].

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

## Acknowledgements

We thank the useful discussions with and feedback from Andrew Tredennick (University of Georgia, USA), Christian Gunning (University of Georgia, USA), Denis Macina (Sanofi Pasteur, France), Laurent Coudeville (Sanofi Pasteur, France), and Edward Thommes (Sanofi Pasteur, Canada). We thank Laura Barrero Guevara (Infectious Disease Epidemiology Group, Max Planck Institute for Infection Biology, Berlin, Germany) for useful feedback on the figures. M.B., E.G., and M.D.C. were funded by the Max Planck Society through the core funding of MDC's Max Planck research group.

## Author contributions

M.B. and M.D.C. designed the model. M.B. performed the simulations. M.B. and M.D.C. analyzed the model output. M.B. performed the systematic review and wrote the first version of the manuscript. M.B. and M.D.C. designed the research. M.D.C. supervised the analyses. M.B., M.D.C., P.R., T.B., and E.G. interpreted the results and made revisions to the paper.

## Funding

## Competing interests

P.R. received funding from Sanofi, France, for another separate project. M.D.C. received post-doctoral funding from Pfizer (2017–2019) and consulting fees from GSK. The remaining authors declare no competing interests.
