## [Peer Review File · Nature Communications]

Maternal pertussis immunization and the blunting of routine vaccine effectiveness: A meta-analysis and modeling studyREVIEWER COMMENTS

Reviewer #1 (Remarks to the Author):

This modeling study evaluated the impact of maternal immunization against pertussis in pregnancy with varying degrees of blunting on pertussis transmission.

This study is important, confirms again the protection provided to young infants by immunization in pregnancy against pertussis, but also the possible increase in pertussis in older infants in years to come as a result of lower active immune responses to infants' own vaccines.

The study still needs to be interpreted cautiously as modeling studies may not prove to be true in the future and these results must be supported by epidemiology data in the years to come.

I have specific comments as below, but the study needs also to take into account the effect of COVID19 and mitigation measures on the transmission of pertussis as well vaccine uptake and pertussis immunity. Studies have shown that COVID-19 and implementation of pharmaceutical interventions led to a decrease in pertussis circulation and immunity (Please see (PMID: 35220973; PMID: 36283895; PMID: 35599039 for example). Do the authors think this may affect the results presented?, would sensitivity analysis be required in the paper to show this?

Specific comments:

Title:

The title needs to be changed in order to better reflect the study question, design and results. It is not clear from the current title.

Short abstract:

Lines 15: Please remove "recently" as this strategy now has a decade experience

Line 18: we do know the clinical and epidemiological consequences, please add long-term as this is what is this study trying to explore

Lines 21-22: "Because of this transient phase, current empirical studies may indicate a 10–20% reduction in infant vaccine effectiveness." This is not clear , please rephrase and be clearer

Line 25- please remove "and other vaccine-preventable diseases" as the focus of the study is pertussis

Long Abstract

Line 303-1- please rephrase the "vulnerable demographic", same applies to short abstract

Line 35: again , the long-term

Lines 39-40 "with some estimates consistent

with blunting.": I do not know what article paper refer to here as I am not award of studies showing clinical effect of blunting

Line 49: "10–20% reduction in vaccine effectiveness"- which age group and which vaccine (ante-natal or post-natal)

Line 51: "Ultimately"- do you mean Third?

Full text:

General comment: Paper needs professional english editing

Line 70 please remove recently as this strategy now has more than a decade experience

Line 86: please note that ref #17 is meta-analysis not in the context of immunization in pregnancy

Lines 87-92: this statement is true for SPN but not for Hib (please see figures and results of MA by Abu-Raya et al , Front. Immunol. 12:689394. doi: 10.3389/fimmu.2021.689394)

Line 95: "only recently" is not entirely true

Line 97-98: Please see now 10 8 years experience in the US (Skoff et al, PMID: 36745442). Is this still short-term ?

Line 108: Is "honeymoon" the scientifically correct term to use ?

Line 127: what was the search criteria ?

Line 130: "having received their primary series" or "having received at least one dose of their primary series"

143: please explain what is meant by failure in degree (or leakiness)?

Line 148: why the authors choose 3 months as in most countries primary immunization occurs at age of 2 months? will this have effect on the model ?

Line 151: what is meant by "mothers whose immunization failed"

Lines 165-184: Did the authors consider adding another parameter of timing of vaccination in pregnancy as it affects levels of antibodies achieved at delivery, although it does not affect blunting (please see PMID: 34598822 and PMID: 37080224). Please comment on this and discuss it in the discussion as it is important point.

Line 184: why the authors choose range of 0-20% of blunting? Levels of anti-pertussis antibodies in infants of vaccinated vs unvaccinated after primary immunization were shown to be in the range of 0.41-

0.68 (please see Fig 2 in PMID: 34305922).

Line 195: please explain abbreviation of PPV?

Line 204: 5% do you mean 0.05 (so it is consistent with the interpretation of RR in lines 197-202)

Line 258: "lasting at least 5 years and followed by a rebound"- this has not happened in any country implementing maternal immunization for more than 5 years. Please explain

Lines 259-261: if the incidence remained lower than before maternal immunization so why that is blunting, to me it is not consistent with blunting but rather added protection via maternal immunization in this age group

Lines 332-335: this is interesting- is this because of effect of antibodies as the main driver of blunting (as in PMID: 34598822)

Lines 343-5: would the authors consider adding PMID: 36745442.

Lines 368-369: pharmaceutical industries are not the bodies recommend vaccination in pregnancy

Lines 370-373: this sentence is not accurate. There are no vaccines trialed in pregnant women for TB, mumps and rotavirus. Please restrict this to pertussis and streptococcus pneumoniae where we have evidence

381-384: it is important to report the meta-analysis showing that the higher antibody levels at primary immunization the lower the immune responses suggesting delaying primary vaccination (see PMID: 34598822), and also that other factors did not affect infants' immune responses

Line 415: please see my comment above on the US study with 8 years experience

Lines 427-429: How COVID-19 will impact the results of this modelling study given that it has been shown that antibody levels and reported incidence rates have decreased during COVID-19 pandemic (see PMID: 35599039)

Dr. Bahaa Abu-Raya

Reviewer #2 (Remarks to the Author):

Authors presented their mathematical modeling study whether maternal immunization blunt the

effectiveness of pertussis vaccines in infants based on a previously developed pertussis transmission model which was empirically validated on data from Massachusetts, USA. This is a very interesting study showing a potential risk of pertussis resurgence due to the impact of the maternal pertussis vaccination programs on paediatric primary pertussis vaccine programs.

My main concern on this study is that the model description was not clear enough to follow easily and if the current model structure is detailed-enough to distinguish the impact between maternal vaccination blunting on primary vaccination programs among infants.

Comments:

Line 73.

In 2015, the World Health Organization (WHO) issued an official recommendation for maternal immunization against pertussis for certain circumstances as mentioned in the 2015 WHO Pertussis Position paper (8) that 1. Conclusions on maternal immunization therefore cannot be extrapolated to infants receiving wP vaccines in the absence of additional immunogenicity and safety data. 2. A switch from wP to aP vaccines for the primary schedule should only be considered if additional periodic booster or maternal immunization can be assured and sustained. National programs currently administering wP vaccination should continue to use wP vaccines for primary vaccination series. National programs currently using aP vaccine may continue using this vaccine but should consider the need for additional booster doses and additional strategies such as maternal immunization in case of resurgence of pertussis.

Line 145.

Authors mentioned that the model follows individuals according to their immunization history. Could you explain more how you trace individuals in your compartmental model structure?

Model structure in Supplementary

Fig S1 described that the model time step is one day but the model ages each age group by proportions. Doesn't it cause the risk of moving vaccinated individuals too fast to the next age groups?

It is not clear if you vaccinated individuals single or multiple times (p_1) with primary schedule vaccines in S1 and the timing of the primary schedules.

In the first age group in S 1.2 equations, the latents (E) are from four susceptible compartments according to λ but the equations for Ss show all zeros (no movement) except for S1. Isn't it an error?

What is the difference between vaccine-failed mothers, and vaccine successful or unvaccinated? Do you assume vaccine-failed mothers still have some protection against pertussis? Do you need these compartments? According to Fig S1, there is no difference between S_m and S. ϵ_M could be assumed zero and p_0 to be an effective vaccine coverage to make the model simpler.

Line 155. Assuming 0.011 per year as the vaccine protection waning parameter means that the average

duration of the vaccine protection is almost 90 years. What is the impact of this long vaccine protection when the model population is limited to 75 years old.

POLYMOD contact

Having the model with the importance of infants in the population, isn't it better to have 0 year old separate in 0-4 Y age group in the POLYMOD age groups?

How do M1 move from the first age group to M2 in the second age group according to $\bar{\epsilon}$? When they get the primary dose, do they get aged at the same time? This is not clearly described in Fig S1.

Line 180.

Fig S2 is very difficult to follow. The x-axis shows the time since the maternal vaccination and the y-axis for the vaccine effectiveness at time. When the average duration is infinite, the VE should not be reduced. Fig S2 suggest the average duration seems much longer than written on the figure. They never reach zero. I would read Fig S2 as Take * Waning.

Line 184. This parameter b_1 is not clearly formulised with $\bar{\epsilon}$. It would be better to show the difference ϵ and $\bar{\epsilon}$ in Table S3. Regarding the prementioned question on multiple primary doses, how does the model deal with blunting on different doses?

Line 234.

When the UK and California studies suggest no blunting with the maternal pertussis vaccination, why would you use their data for this RR comparison.

Line 246.

Nordic countries with high pertussis vaccine coverage have not experienced the resurgence of pertussis while many countries did. Do you think they will have a resurgence based on your model results?

There seems to be a strong correlation between the duration of the maternal vaccine protection and the timing of resurgence due to blunting. Have you examined the sensitivity of this protection duration in your model?

Line 297.

With the RR values, how do you assume an increasing VE of primary doses?

Line 300.

Why do the pre-maternal vaccination era incidences in Fig 5A-C have different incidences with different blunting levels? Blunting should not have any influence before the introduction of maternal vaccination program?

Line 309.

Model results indicate an increase of the pertussis cases among adults after maternal program started. Have you seen any data showing this phenomenon? Has the model included the vaccination impact of the mothers?

It is not easy to distinguish the effectiveness of the maternal and primary vaccines as those with maternal vaccinated mothers are likely to vaccinate their infants.

Line 324.

The coverage of the maternal immunization would influence the impact of the blunting on infant vaccination.

In Fig S6 B&E, I would expect there is no RR change with blunting = zero.

Line 326.

I thought the average duration of the maternal protection was fixed at 8.7 months.

0 year old mixing is very important. need to mention here about that. What is the take for having similar figures with 7 different contact surveys here?

Reviewer #3 (Remarks to the Author):

This study addressed the impact of maternal pertussis immunization on the risk of diseases in infants using 2 methods: a literature review and a modelling component. While this topic is of interest, there are significant methodological concerns.

Abstract

Both the short and long abstract do not provide sufficient details on the methods (both the modelling part and review of epidemiological evidence) that were employed in the study, which immune markers were assessed, against endpoints, which study population, and do not provide specific results such as estimated vaccine effectiveness.

Introduction

The authors use the word 'demographic' to describe 'age group, for example in line 66 'Newborns are the demographic most vulnerable to pertussis'. I suggest using the term 'age group' instead of demographic.

Much of the significant evidence on pertussis maternal immunization was generated initially in the UK, but very few studies were cited. Hallmark trials should be cited in the introduction. Below a list of relevant articles on this topic.

Some epidemiological terms are incorrectly used. For example, in lines 77-78 the authors stated that 'estimates of reductions in pertussis incidence ranging from 70 to 95% [10-14]', while these are reductions in the risk of the disease, rather than reductions in the incidence.

Lines 80-86: the authors should provide more specific details on the impact of maternal pertussis immunization on pertussis immune response in infants from vaccinated mothers.

The rationale of the study is not defined well.

A clearer rephrasing of the study objectives is needed.

Lines 99-103 – these are methods/results.

Lines 106-108 start with 'Our study shows... and end with reference '[24]'. It is not clear, are these results of the current study? Why reference 24 is cited, if these are results of the current analysis.

Lines 106-117 these are results/discussion/conclusions and recommendations, that should be placed in the relevant sections rather than in the introduction (which should set the rational/background of the study).

Methods

The authors conducted a narrative review rather than a systematic review. Narrative reviews are subject to biases and this is a major limitation of this manuscript. Systematic reviews on the other hand are done in a structured manner and less prone to bias, while providing critical methodical details, which allow replication of the study.

It is not clear why the authors conducted a review of studies that assessed the risk of pertussis in infants in relation to maternal immunization without a meta-analysis, which would have much more impact than a review.

In their review the authors refer to heterogeneity between studies, but not formal test was cited in the methods section.

What were the keywords that were used in the literature search?

The authors used Google scholar, but what about the Pubmed and Scopus? Why not including these databases in their literature search?

Please see the following reference: Page M J, McKenzie J E, Bossuyt P M, Boutron I, Hoffmann T C, Mulrow C D et al. The PRISMA 2020 statement: an updated guideline for reporting systematic reviews BMJ 2021; 372 :n71 doi:10.1136/bmj.n71.

Model parametrization: I am concerned that the model parametrization was based on one study (reference# 3) using data from Massachusetts. what is the validity of the approach and potential generalizability of the findings.

There are other approaches, ranked at a higher level of evidence (e.g., cohort studies, case-control studies) for estimating vaccine effectiveness than the 'screening method'. The rational for using this approach is not clear.

The model assumptions of 95% coverage of the primary series and booster vaccination might be too optimistic.

Relevant references:

- 1: Grassly NC, Andrews N, Cooper G, Stephens L, Waight P, Jones CE, Heath PT, Calvert A, Southern J, Martin J, Miller E. Effect of maternal immunisation with multivalent vaccines containing inactivated poliovirus vaccine (IPV) on infant IPV immune response: A phase 4, multi-centre randomised trial. *Vaccine*. 2023 Feb 10;41(7):1299-1302. doi: 10.1016/j.vaccine.2023.01.035.
- 2: Sapuan S, Andrews N, Hallis B, Hole L, Jones CE, Matheson M, Miller E, Snape MD, Heath PT. An observational, cohort, multi-centre, open label phase IV extension study comparing preschool DTAP-IPV booster vaccine responses in children whose mothers were randomised to one of two pertussis-containing vaccines or received no pertussis-containing vaccine in pregnancy in England. *Vaccine*. 2022 Nov 22;40(49):7050-7056. doi: 10.1016/j.vaccine.2022.10.005.
- 3: Oguti B, Ali A, Andrews N, Barug D, Anh Dang D, Halperin SA, Thu Hoang HT, Holder B, Kampmann B, Kazi AM, Langley JM, Leuridan E, Madavan N, Maertens K, Maldonado H, Miller E, Munoz-Rivas FM, Omer SB, Pollard AJ, Rice TF, Rots N, Sundaram ME, Wanlapakorn N, Voysey M. The half-life of maternal transplacental antibodies against diphtheria, tetanus, and pertussis in infants: an individual participant data meta-analysis. *Vaccine*. 2022;40(3):450-458. doi: 10.1016/j.vaccine.2021.12.007.
- 4: Jones CE, Calvert A, Southern J, Matheson M, Andrews N, Khalil A, Cuthbertson H, Hallis B, England A, Heath PT, Miller E. A phase IV, multi-centre, randomized clinical trial comparing two pertussis-containing vaccines in pregnant women in England and vaccine responses in their infants. *BMC Med*. 2021;19(1):138. doi: 10.1186/s12916-021-02005-5
- 5: Amirthalingam G, Campbell H, Ribeiro S, Fry NK, Ramsay M, Miller E, Andrews N. Sustained Effectiveness of the Maternal Pertussis Immunization Program in England 3 Years Following Introduction. *Clin Infect Dis*. 2016 Dec 1;63(suppl4):S236-S243. doi: 10.1093/cid/ciw559.
- 6: Choi YH, Campbell H, Amirthalingam G, van Hoek AJ, Miller E. Investigating the pertussis resurgence in England and Wales, and options for future control. *BMC Med*. 2016 Sep 1;14(1):121. doi: 10.1186/s12916-016-0665-8.
- 7: Kent A, Ladhani SN, Andrews NJ, Matheson M, England A, Miller E, Heath PT; PUNS study group. Pertussis Antibody Concentrations in Infants Born Prematurely to Mothers Vaccinated in Pregnancy. *Pediatrics*. 2016 Jul;138(1):e20153854. doi: 10.1542/peds.2015-3854
- 8: van Hoek AJ, Campbell H, Amirthalingam G, Andrews N, Miller E. Cost-effectiveness and programmatic benefits of maternal vaccination against pertussis in England. *J Infect*. 2016 Jul;73(1):28-37. doi:10.1016/j.jinf.2016.04.012.
- 9: Ladhani SN, Andrews NJ, Southern J, Jones CE, Amirthalingam G, Waight PA, England A, Matheson M, Bai X, Findlow H, Burbidge P, Thalasselis V, Hallis B, Goldblatt D, Borrow R, Heath PT, Miller E. Antibody responses after primary immunization in infants born to women receiving a pertussis-containing vaccine during pregnancy: single arm observational study with a historical comparator. *Clin Infect Dis*. 2015 Dec 1;61(11):1637-44. doi: 10.1093/cid/civ695.
- 10: Amirthalingam G, Andrews N, Campbell H, Ribeiro S, Kara E, Donegan K, Fry NK, Miller E, Ramsay M. Effectiveness of maternal pertussis vaccination in England: an observational study. *Lancet*. 2014 Oct 25;384(9953):1521-8. doi:10.1016/S0140-6736(14)60686-3.

REVIEWER COMMENTS

Reviewer #1 (Remarks to the Author):

This modeling study evaluated the impact of maternal immunization against pertussis in pregnancy with varying degrees of blunting on pertussis transmission. This study is important, confirms again the protection provided to young infants by immunization in pregnancy against pertussis, but also the possible increase in pertussis in older infants in years to come as a result of lower active immune responses to infants' own vaccines. The study still needs to be interpreted cautiously as modeling studies may not prove to be true in the future and these results must be supported by epidemiology data in the years to come.

Thank you.

I have specific comments as below, but the study needs also to take into account the effect of COVID19 and mitigation measures on the transmission of pertussis as well vaccine uptake and pertussis immunity. Studies have shown that COVID-19 and implementation of pharmaceutical interventions led to a decrease in pertussis circulation and immunity (Please see (PMID: 35220973; PMID: 36283895; PMID: 35599039 for example). Do the authors think this may affect the results presented? Would sensitivity analysis be required in the paper to show this?

We thank the reviewer for emphasizing this point. In the first submission, we limited this point to a few sentences in the discussion section of the manuscript. In the revised manuscript, we follow the reviewer's suggestion in more detail and address this point more thoroughly using simulations in sensitivity analysis 9.

As the reviewer mentions, social distancing measures decrease vaccine uptake and transmission. Hence, we performed a sensitivity analysis following an approach used in an earlier study (Brett & Rohani 2020). First, we simulated a 3 to 9-month decrease in transmission (=social distancing) of 20%, 50%, and 70%, and second, we added a 3-month reduction in vaccination coverage of 20% for all vaccines (Causey et al. 2021, Kim et al. 2022, Lai et al. 2023). Social distancing produced the expected changes in pertussis incidence: a reduction in pertussis incidence during the social distancing period, followed by a pertussis increase once distancing is stopped (e.g., large changes in between Fig. 4 and Fig. S34 A-D). In terms of estimating the impact of maternal immunization, surprisingly, social distancing had no impact on the relative risk of pertussis in infants from vaccinated mothers relative to unvaccinated mothers (e.g., no changes between Fig. 5 and Fig. S34 E & F).

We have added the results in the results section (section sensitivity analysis ix, l. 431 and l. 447) and in the supplementary information, pages 31–40 (see illustration of one sensitivity analysis below). We explain the methods for this sensitivity analysis in the methods section l. 847.

Fig. S34: Sensitivity analysis 9: RR of pertussis in infants from vaccinated mothers relative to unvaccinated mothers with social distancing that has a 20% reduction in both transmission and coverage of all immunizations. Panels G and H compare this social distancing scenario relative to a scenario without social distancing (i.e., as in Fig. 4 and Fig. 5). In panels G and H, we selected, for illustration purposes, 10% blunting for both with and without social distancing. In panel (G), the variation around the first diagonal reflects stochasticity between simulations.

Specific comments:

Title:

The title needs to be changed in order to better reflect the study question, design and results. It is not clear from the current title.

Thanks for this excellent suggestion. We have replaced the title with: 'Does maternal immunization blunt the effectiveness of pertussis vaccines in infants? A systematic review and modeling study'

Short abstract:

Lines 15: Please remove "recently" as this strategy now has a decade experience

We have now replaced the sentence by (l. 17): "In the last decade, maternal immunization has been deployed in many countries, successfully reducing pertussis in this age group."

Line 18: we do know the clinical and epidemiological consequences, please add long-term as this is what is this study trying to explore

In the revised abstract, we have deleted this sentence.

Lines 21-22: "Because of this transient phase, current empirical studies may indicate a 10–20% reduction in infant vaccine effectiveness." This is not clear, please rephrase and be clearer

We have now rephrased this sentence as (l. 25): "We show that transient dynamics can mask blunting for at least a decade after rolling out maternal immunization."

Line 25- please remove "and other vaccine-preventable diseases" as the focus of the study is pertussis

We have removed this as suggested (l. 27).

Long Abstract

Line 303-1- please rephrase the "vulnerable demographic", same applies to short abstract

Line 35: again, the long-term

Lines 39-40 "with some estimates consistent with blunting.": I do not know what article paper refer to here as I am not aware of studies showing clinical effect of blunting

Line 49: "10–20% reduction in vaccine effectiveness"- which age group and which vaccine (ante-natal or post-natal)

Line 51: "Ultimately"- do you mean Third?

Thank you for these comments. Following the journal formatting instructions, we have removed the long abstract.

Full text:

General comment: Paper needs professional english editing

We appreciate the reviewer's comment, but we respectfully point out that the manuscript has been thoroughly checked by three native English-speaking co-authors (EG, TB, and PR) and further entirely proofread with Grammarly, a state-of-the-art typing assistant.

Line 70 please remove recently as this strategy now has more than a decade experience

We have removed 'recently', and we now use the year of first introduction, i.e., 2012. The sentence now reads as (l. 170): '...since 2012, many countries have introduced maternal immunization, ...'

Line 86: please note that ref #17 is meta-analysis not in the context of immunization in pregnancy

Thank you for pointing out: indeed, ref 17 compares infants with pre-existing maternal antibodies relative to those without, and hence we have excluded it after this sentence.

Lines 87-92: this statement is true for SPN but not for Hib (please see figures and results of MA by Abu-Raya et al, Front. Immunol. 12:689394. doi: 10.3389/fimmu.2021.689394) Thank you for pointing out your meta-analysis on this, we have now excluded Hib from the sentence. We refer to your 2021 Front. Immunol. meta-analysis here (l. 201) and elsewhere in the manuscript (e.g., l. 186).

Line 95: "only recently" is not entirely true

We have replaced 'only recently' by '2012', and the sentence now reads as (l. 203): 'While immunological blunting is well documented, because the first maternal immunization programs were implemented in 2012, the long-term consequences of blunting on vaccine effectiveness (VE) and the ensuing epidemiology of pertussis remain difficult to evaluate.'

Line 97-98: Please see now 10: 8 years experience in the US (Skoff et al, PMID: 36745442). Is this still short-term ?

We very much appreciate the excellent work by Dr. Tami Skoff and their team and we cite several of their publications in our study. Unfortunately, this particular study did not make it through our inclusion criteria because it is an era-based comparison instead of an individual-participant study. It does not estimate RR, but it provides useful information on maternal immunization coverage and age-specific pertussis incidence rates (Skoff et al. 2023, Fig. 1), and hence we cite it in the discussion.

That being said, in the revision, we have deleted the sentence that triggered this comment. To avoid the discussion on what constitutes a short vs. long-term study, we have revised the manuscript to avoid comments on 'short-term'. Now, we state that 2012 was the start of the first maternal immunization programs, and we clarify that the purpose of the simulation is to identify the population-level effect of blunting and maternal immunization over several decades. We hope that this approach better respects the efforts made by the researchers who monitor the impact of maternal immunization. L. 211 reads as: "Second, we extend a previously validated model of pertussis immunization^{4,27} to simulate over several decades the short- and long-term epidemiological impact of maternal immunization with various levels of blunting and maternal immunization coverages on the age-specific time series of pertussis incidence."

Line 108: Is "honeymoon" the scientifically correct term to use ?

A honeymoon effect is quite a common term in epidemiology. It was originally coined by McLean & Anderson in 1988. It is defined as 'the deep trough in the number of infections that accompanies the sudden introduction of vaccination'. It is also explained in epidemiology textbooks, for example by Keeling & Rohani 2008 (p. 295) and in various review papers and manuscripts on immunization programs, e.g., Ahmed et al. 2007, Riolo et al. 2013, Domenech de Cellès et al. 2018, Metcalf et al. 2020.

Line 127: what was the search criteria ?

Following the comments of reviewer 3, we have repeated the search and rewritten this section. We explain the revision in detail in the rebuttal to reviewer 3, but in brief, the criteria were (l. 675): “We searched the literature on Web of Science, PubMed, and Scopus on August 25, 2023. We used the search terms “maternal immunization AND pertussis AND effectiveness OR maternal vaccination AND pertussis AND effectiveness”. In Web of Science, we entered these terms under ‘Topic’, in PubMed we entered these terms under ‘Title/Abstract’, and in Scopus under ‘Article type, Abstract, Keywords’. In these three searches, we selected the publication dates from January 1, 2012, which is the first complete year with recommended maternal immunization in any country¹⁴, until August 25, 2023.”

Line 130: "having received their primary series" or "having received at least one dose of their primary series"

Thank you for pointing this out, we have adjusted this to (l. 683): ‘...having received at least one dose of their primary immunization series...’

143: please explain what is meant by failure in degree (or leakiness)?

We have revised this sentence and explain the leakiness as (l. 737): ‘Previous results in⁴ found no evidence for failure in degree (or leakiness, i.e., when vaccine-induced protection is imperfect and vaccinees remain susceptible to infection, but at a lower degree than unvaccinated individuals), and hence we ignored this possibility.’

Line 148: why the authors choose 3 months as in most countries primary immunization occurs at age of 2 months? will this have effect on the model ?

In sensitivity analysis 3, we showed the impact of the age at the start of the primary series, varying from 2 months until 9 months, on blunting and pertussis incidence. The difference between starting the primary series at two months or three months is rather small, but the earlier the start of the primary series, the larger the blunting effect on the RR (Fig S11, S12).

We chose to present a start of the primary series at 3 months for two reasons: (i) while the recommended age is two months, in reality, between 5% and 35% of children have a 1-month delay in immunization (e.g., Riise 2015, Rane et al. 2021), and (ii) while our take home-messages become more pronounced when starting the primary series at 2 months, we wanted to avoid ‘cherry-picking’ an immunization schedule that gave the most pronounced results.

Line 151: what is meant by “mothers whose immunization failed”

These are mothers who received maternal immunization during pregnancy, but whose newborn remained unprotected because pertussis vaccines are not 100% effective. In our model, the effectiveness of maternal immunization is based on several epidemiological studies, including for example, Darbrera et al. 2015 (VE= 93%, 95% CI=81%–97%), Skoff et al. 2017 (77.7%, 95% CI= 48.3%–90.4%, or 83.0% (95% CI=49.6%–94.3%), or Amirthalingham et al. (89%, 95% CI=86%–91%) or the review provided by Kandeil et al. 2020, figure 5. Serological studies in mothers reported similar VE estimates, for example, 88%–93% of pregnant women had increased antibodies titers against pertussis antigens 1 month after vaccine administration relative to pre-vaccine concentrations (Perret et al. 2020, Table 3). We clarify this concept as follows (l. 754): ‘mothers whose immunization failed (i.e., who received the vaccine but whose infant remained unprotected),’

Lines 165-184: Did the authors consider adding another parameter of timing of vaccination in pregnancy as it affects levels of antibodies achieved at delivery, although it does not affect blunting (please see PMID: 34598822 and PMID: 37080224). Please comment on this and discuss it in the discussion as it is important point.

In the revision, we have added this point in the methods section, l. 755: We model maternal immunization by transferring a fraction of newborns to a protected class. Maternal immunization in itself is not modeled; hence our model does not specify at which stage of pregnancy maternal immunization occurs. The assumed high effectiveness of maternal immunization implies that it occurred at a timing consistent with that used in the studies in table S1 and in the empirical studies of the systematic review, i.e., during the second or third trimester of pregnancy and at least one week before giving birth (Table S1).

Line 184: why the authors choose range of 0-20% of blunting? Levels of anti-pertussis antibodies in infants of vaccinated vs unvaccinated after primary immunization were shown to be in the range of 0.41-0.68 (please see Fig 2 in PMID: 34305922).

The reviewer raises an important point. We have chosen this range because it best fits with the estimates of RR derived from empirical studies. As your meta-analysis clearly shows, the levels of pertussis antibodies are between 30% and 60% lower in infants from vaccinated vs. unvaccinated mothers, but unfortunately, we do not know how such changes in antibody titers translate into protection.

In the first submission, we mentioned this rationale on page 2 of the supplementary information. Following your comment, we have moved this to the methods section l. 808: "There are no empirical estimates of the degree of blunting of infant protection following maternal immunization or how changes in IgG titers translate into changes in protection. When infants receive their primary immunization, the antibody concentrations of infants from vaccinated mothers are 30% to 60% lower compared to those from unvaccinated mothers^{21,22}. To best match the empirical estimates of the relative risk of pertussis, we chose a decrease in the effectiveness of primary immunization from 0% (i.e., no blunting) to 20%. Formally, the blunting parameter b_1 is modeled as a relative reduction in the initial effectiveness of infant immunization e , resulting in a blunted effectiveness."

To support the choice of the levels of blunting, we conducted simulations with blunting of the effectiveness of the primary series up to 60%. These simulations reinforce our main take-home messages, namely: (i) that a few years of epidemiological monitoring result in unreliable estimates of vaccine effectiveness and blunting, and (ii) that according to our model, we can exclude high levels of blunting based on one study with 6 years of monitoring. We have added this sensitivity analysis to the results section (sensitivity analysis vii, l. 441), to the methods section (sensitivity analysis vii, l. 844), and to the supplementary information, p. 27 & 28. We show this sensitivity analysis here below.

Fig S21: Sensitivity analysis 7: Pertussis incidence before and after the start of the maternal immunization program without blunting (0%) and with blunting levels up to 60%. All other parameter values are as in Fig. 4, which shows scenarios with a maximum of 20% blunting.

Fig. S22: Sensitivity analysis 7: The RR of pertussis infection in infants from vaccinated relative to unvaccinated mothers without blunting (0% blunting) and with blunting levels up to 60%. All other parameter values are as in Fig. 5, which shows scenarios with a maximum of 20% blunting.

Line 195: please explain abbreviation of PPV?

PPV is the proportion of population vaccinated, a quantity equivalent to vaccine coverage that enters the calculation of vaccine effectiveness for the screening method. We have rephrased the sentence l. 873 as: '... where PCV (proportion of cases vaccinated) is the proportion of cases in vaccinated infants born to vaccinated mothers among all cases in infants vaccinated, and PPV is the proportion of the population vaccinated, here equal to the vaccine coverage of maternal immunization.'

Line 204: 5% do you mean 0.05 (so it is consistent with the interpretation of RR in lines 197-202

Yes, thank you. We have changed the sentence so that it matches the interpretation of RR and l. 876 now reads as: "We validated the RR estimates of the simulations by setting the half-life of maternally derived protection to infinity, which resulted in a RR of 0.05 in newborns and matched the initial 95% effectiveness of maternal immunization against pertussis in unvaccinated newborns fixed in the simulations (Fig. S2, Table S3)."

Line 258: "lasting at least 5 years and followed by a rebound"- this has not happened in any country implementing maternal immunization for more than 5 years. Please explain

This is a good point, which can have several causes. In the discussion section of the revised manuscript, we now devote a new paragraph to this topic (l. 581). In brief, the take-home message of this paragraph is that if there are any age-specific blunting-mediated increases in incidence, they will likely go undetected because (i) they are small and occur simultaneously with large decreases in pertussis incidence mediated by maternal immunization (i.e., the increase is relative to a scenario without blunting, not relative to a scenario without maternal immunization), (ii) studies should correct for age-specific temporal trends in pertussis incidence before the roll-out of maternal immunization and (iii) the interpretation of trends in pertussis incidence may be further compounded by changes in national immunization schedules (e.g.,⁴⁰) and the impact of NPIs during the COVID-19 pandemic. Given these three challenges, a rebound in pertussis incidence, if any, is likely difficult to detect. This explains our choice to focus on RR as an endpoint, a metric that is more robust as it allows control for the above-mentioned confounders in empirical studies.

Lines 259-261: if the incidence remained lower than before maternal immunization so why that is blunting, to me it is not consistent with blunting but rather added protection via maternal immunization in this age group

It is correct that the introduction of maternal immunization (both with and without blunting) decreases pertussis incidence in this age group, hence maternal immunization is beneficial. However, in order to understand the effect of blunting on pertussis incidence, we need to compare maternal immunization without blunting relative to maternal immunization with blunting: maternal immunization without blunting is more beneficial than maternal immunization with blunting.

In the revision, we clarify this sentence as follows in l. 349: "In newborns, pertussis incidence was lower after the start of maternal immunization relative to before, but the

benefit of maternal immunization was larger in a scenario without blunting than in a scenario with blunting (Fig. 4A)."

Lines 332-335: this is interesting- is this because of effect of antibodies as the main driver of blunting (as in PMID: 34598822)

Thank you. Indeed, the fast waning of maternal antibodies drives the decreasing blunting effect with later maternal immunization. Now we explain this further and we added the reference in the discussion section, l. 514: "Given the fast waning of maternal antibodies and the negative association between the maternal antibody titers and the blunting of infant immune responses³⁸, delaying infant primary immunization by a few months might greatly reduce the maternal blunting of infant immunization³⁶."

Lines 343-5: would the authors consider adding PMID: 36745442.

In the revision, we added this reference elsewhere in the discussion, namely l. 582: "Epidemiological studies have, until now, not reported maternal immunization-mediated rebounds in pertussis incidence (e.g., Skoff et al. 2023)."

Lines 368-369: pharmaceutical industries are not the bodies recommend vaccination in pregnancy

This is exactly right. We have replaced the sentence with (l. 493): and the pharmaceutical industry is developing new vaccines for immunization during pregnancy.

Lines 370-373: this sentence is not accurate. There are no vaccines trialed in pregnant women for TB, mumps and rotavirus. Please restrict this to pertussis and streptococcus pneumoniae where we have evidence

This sentence was referring to blunting rather than vaccine development. To avoid confusion, we have rephrased this sentence as (l. 495): "In addition to the Tdap and RSV vaccines, maternal immunization is recommended or under development for influenza, COVID-19, polio and several other infections³¹⁻³³"

381-384: it is important to report the meta-analysis showing that the higher antibody levels at primary immunization the lower the immune responses suggesting delaying primary vaccination (see PMID: 34598822), and also that other factors did not affect infants' immune responses

We have added the reference here (l. 513): "Given the fast waning of maternal antibodies and the negative association between the maternal antibody titers and the blunting of infant immune responses³⁸, delaying infant primary immunization by a few months might greatly reduce the maternal blunting of infant immunization³⁶."

Line 415: please see my comment above on the US study with 8 years experience

Thank you for bringing this study to our attention, which we cite elsewhere. However, since it is not an individual-based study that estimates relative risks, it cannot be used to

estimate relative risks or blunting. We have rephrased our sentence l. 562 as: ‘...the current individual-based epidemiological studies had ≤ 6 years of monitoring...

Lines 427-429: How COVID-19 will impact the results of this modelling study given that it has been shown that antibody levels and reported incidence rates have decreased during COVID-19 pandemic (see PMID: 35599039)

We address this point in the first comment of the reviewer. We have carried out new simulations with social-distancing measures that reproduce the observed decreases in pertussis incidence during the COVID-19 pandemic. Our simulations predict that after social distancing measures are lifted, there will be pertussis outbreaks, a phenomenon currently observed in several high-income countries (e.g., in Denmark, Dalby & Andersen, Statens Serum Institut, personal communication). In contrast, the time series of RR remains unaffected, which strengthens our conclusion on the transient dynamics and the need for individual-based RR studies that continue monitoring beyond the currently available studies.

Dr. Bahaa Abu-Raya

Reviewer #2 (Remarks to the Author):

Authors presented their mathematical modeling study whether maternal immunization blunt the effectiveness of pertussis vaccines in infants based on a previously developed pertussis transmission model which was empirically validated on data from Massachusetts, USA. This is a very interesting study showing a potential risk of pertussis resurgence due to the impact of the maternal pertussis vaccination programs on paediatric primary pertussis vaccine programs.

Thank you.

My main concern on this study is that the model description was not clear enough to follow easily and if the current model structure is detailed-enough to distinguish the impact between maternal vaccination blunting on primary vaccination programs among infants.

We have included all your comments regarding the clarity of the model explanation, both in the main text and in the Supplementary Information, which we explain one by one below.

Regarding your concern: In brief, we are able to distinguish between maternal vaccination and blunting because we model maternal immunization programs both with and without blunting (i.e., comparing the gradient red colored lines in Fig. 4 & Fig. 5).

Comments:

Line 73.

In 2015, the World Health Organization (WHO) issued an official recommendation for maternal immunization against pertussis for certain circumstances as mentioned in the 2015 WHO Pertussis Position paper (8) that 1. Conclusions on maternal immunization therefore cannot be extrapolated to infants receiving wP vaccines in the absence of additional immunogenicity and safety data. 2. A switch from wP to aP vaccines for the primary schedule should only be considered if additional periodic booster or maternal immunization can be assured and sustained. National programs currently administering wP vaccination should continue to use wP vaccines for primary vaccination series. National programs currently using aP vaccine may continue using this vaccine but should consider the need for additional booster doses and additional strategies such as maternal immunization in case of resurgence of pertussis.

Thank you for this excellent summary. We have adjusted this sentence, now l. 173 as: "In 2015, the World Health Organization (WHO) issued an official recommendation for maternal immunization with acellular vaccines against pertussis (based on studies with acellular primary immunization)⁹,..."

We believe this sentence matches with what is written in the 2015 WHO report p. 451: "Recent evidence consistently indicates that maternal immunization with aP-containing vaccine during the third trimester of pregnancy is safe and highly effective in protecting infants from pertussis and that it may have a high impact on morbidity and mortality in infants too young to have been vaccinated..."

It also matches with what is written in the 2015 WHO report p. 452: “The evidence reviewed relates only to the use of an aP-containing vaccine in pregnancy, and the immunogenicity data are confined to infants vaccinated with aP-containing vaccines. Conclusions on maternal immunization therefore cannot be extrapolated to infants receiving wP vaccines in the absence of additional immunogenicity and safety data.”

The WHO position on p. 457 states: “Vaccination of pregnant women is likely to be the most cost-effective additional strategy for preventing disease in infants too young to be vaccinated and appears to be more effective and favorable than cocooning. National programmes may consider the vaccination of pregnant women with 1 dose of Tdap (in the 2nd or 3rd trimester and preferably at least 15 days before the end of pregnancy) as a strategy additional to routine primary infant pertussis vaccination in countries or settings with high or increasing infant morbidity/ mortality from pertussis. Cocooning may have an impact on disease prevention in some settings if high coverage can be achieved in a timely manner.”

Line 145.

Authors mentioned that the model follows individuals according to their immunization history. Could you explain more how you trace individuals in your compartmental model structure?

As the reviewer correctly points out, our initial formulation was not completely accurate: Our compartment model allows us to track groups of individuals, not specific individuals. We have rewritten this paragraph such as to better explain how the model follows an individual’s immunization history l. 744: “The model is designed such that we follow individuals grouped according to their immunization history: for each immunization, an individual can go to either of three compartments: one for successful immunization, one for failed immunization, and one for immunization not received (Fig. S1). These three possible compartments start from their mother’s immunization status during pregnancy, followed by an infant immunization schedule that resembles that of the empirical studies in Table S1: the infant’s primary immunization occurs at the age of three months, and an infant booster at the age of 1.5 years. Hence, newborns can be born in three possible compartments: from vaccinated mothers whose immunization succeeded, mothers whose immunization failed (i.e., who received the vaccine but whose infant remained unprotected), or unvaccinated mothers. We model maternal immunization by transferring a fraction of newborns to a protected class. Maternal immunization in itself is not modeled; hence our model does not specify at which stage of pregnancy maternal immunization occurs. The assumed high effectiveness of maternal immunization implies that it occurred at a timing consistent with that used in the studies in Table S1 and in the empirical studies of the systematic review, i.e., during the second or third trimester of pregnancy and at least one week before giving birth.”

Model structure in Supplementary

Fig S1 described that the model time step is one day but the model ages each age group by proportions. Doesn’t it cause the risk of moving vaccinated individuals too fast to the next age groups?

For simplicity, we made the assumption of continuous aging, which is common for age-structured models. Because of the exponentially distributed time of residency in every compartment, this assumption indeed causes a fraction of individuals to age too fast (although the overall demographic structure is preserved). However, this concern should be limited because of the fine age resolution (1-yr age blocks) we used in our model. In a previous modeling study that used a similar age resolution (Domenech de Celles et al. 2019), we found that models with discrete aging (in which individuals age in discrete time every year) produced comparable dynamics.

It is not clear if you vaccinated individuals single or multiple times (p1) with primary schedule vaccines in S1 and the timing of the primary schedules.

The primary immunization is modeled as one event. We clarify this in:

(i) The methods section l. 768: “For simplicity, as in earlier modeling studies⁴, we model the infant primary immunization as one event and we do not consider the gradual effect of individual vaccine doses.”

(ii) The supplementary information, page 2: “We chose a pertussis immunization schedule that closely resembles that of the empirical studies for which we obtained the RR estimates (Table S1). We modeled the infant primary immunization as one event at the age of 3 months, instead of three events at the age of 2,3, and 4 months or 2, 4, and 6 months (Table S1).”

In the first age group in S 1.2 equations, the latents (E) are from four susceptible compartments according to lamda but the equations for Ss show all zeros (no movement) except for S1. Isn't it an error?

This was not an error, just a clarification. Indeed, for the first age group, only one transition into the E compartment was not null, because no one is vaccinated in this group. To avoid confusion, we have now deleted the null transitions from the equation and the equation on page 3 reads as: $\dot{E}_1 = \lambda_1(t)S_1 - (\sigma + \delta_1)E_1$

What is the difference between vaccine-failed mothers, and vaccine successful or unvaccinated? Do you assume vaccine-failed mothers still have some protection against pertussis? Do you need these compartments? According to Fig S1, there is no difference between Sm and S. EpsilonM could be assumed zero and p0 to be an effective vaccine coverage to make the model simpler.

The reviewer is exactly right: there is no difference between Sm and S, so that these two compartments could theoretically be merged and the vaccine coverage replaced with an effective vaccine coverage. However, we separated them for bookkeeping purposes, to be able to distinguish infections in infants (second age group) vaccinated and born to vaccinated mothers (transitions $\mathbb{V}^{(\mathbb{V}\bar{\mathbb{V}})} \rightarrow \mathbb{V}$, class 3 infections in Fig. S1) from those in infants vaccinated and born to unvaccinated mothers (transitions $\mathbb{V}^{(\mathbb{V})} \rightarrow \mathbb{V}$, class 2 infections). Though dynamically neutral, this distinction was essential to compute the model-derived relative risk for subsequent comparison with empirical studies.

Line 155. Assuming 0.011 per year as the vaccine protection waning parameter means that the average duration of the vaccine protection is almost 90 years. What is the impact of this long vaccine protection when the model population is limited to 75 years old.

The reviewer raises an interesting point. In our model, the duration of vaccine protection follows an Exponential distribution, so that, even though the average duration is long, a sizeable proportion of vaccinees lose protection shortly after vaccination. For example, for a rate of 0.011 per year, the proportion losing immunity within 5 years (a metric we used in a previous study) is 5.3% and 10.4% within 10 years. Of note, the fact that we could precisely estimate this rate based on incidence data in Massachusetts (see Domenech de Cellès et al., 2018) demonstrates the long-term dynamical impact of even low rates of waning immunity.

POLYMOD contact

Having the model with the importance of infants in the population, isn't it better to have 0 year old separate in 0-4 Y age group in the POLYMOD age groups?

We agree that a more precise estimation of contact patterns in newborns would be informative. However, neither the original study by Mossong et al. 2008, nor its 2022 implementation in R (Funk & Willem 2022) were powered to provide reliable estimates at this resolution. Instead, they reported contact patterns in 5-yr age blocks, a breakdown also used in most modeling studies based on these data, including ours. Similarly, an often-used follow-up study (Prem et al. 2017), where contact matrices were not based on a survey, but inferred from country-specific demographic data, did not distinguish between these age groups either. We believe that the reviewer has made a very valuable comment, but addressing this problem requires a study on its own, ideally based on a survey, or alternatively inferred from demographic data.

How do M1 move from the first age group to M2 in the second age group according to $\bar{\epsilon}$? When they get the primary dose, do they get aged at the same time? This is not clearly described in Fig S1.

For simplicity, we used a compact model schematic and indeed did not represent the transitions for individuals aging but staying in the same class. In the model, a fraction $1 - \beta_1^{(Q)}$ transitions to β_2 upon aging from β_1 , representing unvaccinated infants born to vaccinated mothers. As the reviewer correctly points out, in our model, vaccination always occurs when aging, so that primary immunization and aging are assumed simultaneous (transitions $\beta_1 \rightarrow \beta_2^{(Q)}$ and $\beta_1 \rightarrow \beta_2^{(Q)}$).

We have clarified this in the legend of Fig. S1, where we have added: "For simplicity, we omitted the transitions between age groups for individuals remaining in the same immunization status, e.g., a fraction $1 - p_1(t)$ transitioning upon aging from M1 to M2."

Line 180.

Fig S2 is very difficult to follow. The x-axis shows the time since the maternal vaccination and the y-axis for the vaccine effectiveness at time. When the average duration is infinite, the VE should not be reduced. Fig S2 suggest the average duration seems much longer than written on the figure. They never reach zero. I would read Fig S2 as Take * Waning.

We see the reviewer's point and we apologize for any confusion. Figure S2 does not actually represent the effectiveness of maternal immunization (against pertussis in unvaccinated newborns, VE) in a cohort of newborns followed over time—in which case, as the reviewer correctly points out, we would expect a larger drop in VE with waning and no drop without waning.

Instead, this figure represents the VE calculated at different time points since the rollout of maternal immunization (that is, in different cohorts of newborns of the same age). Hence, this figure is comparable to Figure 5 in the main text (which represents $RR=1-VE$ in vaccinated infants), and the gradual decrease in VE in Fig. S2 is caused by the same transient effect (in brief, the deleterious impact of blunting takes years to become manifest, until blunted-vaccinated children reach school age). When the duration of maternally-derived antibodies is assumed infinite, Fig. S2 shows that, after the transient phase, the VE estimated from the screening method reaches the nominal value fixed in our model ($1 - \beta_{\text{bl}} = 0.95$). This result thus validates the use of the screening method for estimating VE, in the context of our model.

For the same reasons (i.e., because the figure represents different cohorts with individuals of the same age), in Fig. S2, VE is not expected to reach 0, even though at the individual level, in our model, there is waning of maternal antibodies following an Exponential distribution with the parameter τ^{-1} represents the average duration of protection conferred by the mothers' antibodies to their unvaccinated newborns.

To clarify the figure, we have added an explanation in the legend of Fig. S2: "Population-level validation of the estimation of maternal vaccine effectiveness in newborns between the age of 0 and 2 months, with varying duration of maternally derived immunity. In this figure, the dependent variable represents the VE calculated at different time points since the start of maternal immunization (that is, in different cohorts of newborns between 0 and 2 months and is therefore comparable to Fig. 5 in the main text (which represents $RR=1-VE$ in vaccinated infants). In this figure, the gradual decrease in VE over time is caused by the same transient effect as described in the results section: in brief, the deleterious impact of blunting takes years to become manifest, until blunted-vaccinated children reach school age. When the duration of maternally-derived antibodies is assumed infinite, this figure shows that, after the transient phase, the VE estimated from the screening method reaches the nominal value fixed in our model (0.95). When the average duration of maternally derived immunity is close to 8.7 months, which is the value used in the simulations, we obtain at equilibrium, an effectiveness in newborns close to 80%, which is the estimate obtained in some studies, e.g, Skoff et al. 2017."

Line 184. This parameter b_1 is not clearly formulised with $\bar{\epsilon}$. It would be better to show the difference ϵ and $\bar{\epsilon}$ in Table S3.

In l. 817 we have added: Formally, the blunting parameter β_{bl} is modeled as a relative reduction in the initial effectiveness of infant immunization ϵ , resulting in a blunted effectiveness $\bar{\epsilon} = \epsilon(1 - \beta_{\text{bl}})$. In Table S3, we have added the difference between ϵ and $\bar{\epsilon}$.

Regarding the prementioned question on multiple primary doses, how does the model deal with blunting on different doses?

For simplicity, primary immunization in infants is modeled as one event. Hence, the blunting of the primary immunization is also modeled as occurring during this event. We clarify this in the methods section l. 768: “For simplicity, as in earlier modeling studies⁴, we model the infant primary immunization as one event and we do not consider the gradual effect of multiple vaccine doses.”

Line 234.

When the UK and California studies suggest no blunting with the maternal pertussis vaccination, why would you use their data for this RR comparison.

Following the comment of reviewer 3, we have formalized the empirical part of our study as a systematic review, and hence we give an overview of all estimates that are currently available. Usually, it is considered that because their mean values are around one (ignoring for a moment the large confidence around these estimates), their results do not indicate blunting. Our simulations show that taking into account the transient dynamics of relative risks following the roll-out of an immunization program, the empirical estimates are actually consistent with some degree of blunting.

We clarify this point among others, in the results section and in the discussion. In the results section, we write in l. 388: “During this transient phase, even in scenarios with blunting, the RR was initially far below 1 and gradually increased after the start of maternal immunization. Hence, ignoring these transient dynamics and assuming that a RR of 1 indicates no blunting may result in overestimating the effectiveness of maternal immunization in both unvaccinated newborns and vaccinated infants (Fig. 5B).”

In the discussion section, we write in l. 561: ‘From a practical perspective, the transient phase following maternal immunization has at least two important implications. First, the current individual-based epidemiological studies had ≤ 6 years of monitoring, which we predict is insufficient to capture the consequences of maternal immunization that can only be detected with at least a decade of data. Second, an RR of 1 is considered the baseline value indicating that maternal immunization does not blunt vaccine effectiveness in infants. In the transient phase, however, our model predicts that the baseline value under a no-blunting scenario starts off close to 0 and gradually increases to an equilibrium just below 1.’

Line 246.

Nordic countries with high pertussis vaccine coverage have not experienced the resurgence of pertussis while many countries did. Do you think they will have a resurgence based on your model results?

This is an interesting point. We happen to carry out two projects on pertussis incidence in Sweden and Denmark. As Dalby et al. 2016 pointed out, Denmark is experiencing a rise in pertussis incidence in older age groups (which started before the rollout of maternal immunization).

At the moment, Denmark is the only Nordic country that introduced maternal immunization in 2020. Hence, our study on maternal immunization may only apply to Denmark, and unfortunately, the introduction of maternal immunization coincides with

the COVID-19 pandemic and nationwide social distancing measures such that making predictions is difficult in this case.

We address the possibility of resurgence and an empirical test of our model predictions in the discussion section. In brief, while our model indicates increases in pertussis incidence in some age classes, these remain small relative to the decrease in newborns (Fig. 4) and hence will likely remain undetected. We believe the best empirical test of our model would be to quantify whether there are increases in RR in newborns or infants during the primary series over time with a dataset that contains decent sample sizes. We added a paragraph in the discussion section to explain this in more detail in l. 581.

There seems to be a strong correlation between the duration of the maternal vaccine protection and the timing of resurgence due to blunting. Have you examined the sensitivity of this protection duration in your model?

Thank you for pointing this out. We examined whether our conclusions were sensitive to the duration of maternal vaccine protection (sensitivity analysis iii, Figs. S9 & S10). We ranged the average duration of maternal protection from 4 months to 1 year. Our conclusions were consistent across this variation: Fig S10 B vs. E and Fig S10 C vs. F show that the transient phase lasted at least a decade, and the current empirical estimates of RR in infants from vaccinated mothers relative to unvaccinated mothers are consistent with some minor blunting of infant primary immunization.

Line 297.

With the RR values, how do you assume an increasing VE of primary doses?

In our model, infant primary immunization is one event. Hence, we cannot take into account an increasing VE of primary doses. We assume a VE of the primary immunization based on a previous study (Domenech de Cellès et al. 2018), which was consistent with or close to the empirical estimates of the primary immunization as a whole.

Line 300.

Why do the pre-maternal vaccination era incidences in Fig 5A-C have different incidences with different blunting levels? Blunting should not have any influence before the introduction of maternal vaccination program?

The reviewer is correct that blunting does not have any influence on pertussis incidence before the introduction of the maternal vaccination program. Fig 5 does not show any data before the introduction of the vaccination program (X-axis with time since start of maternal immunization program >1), so we assume that the reviewer is referring to Fig 4. In Fig 4, we show as a control and consistent with the expectation, that the different simulated scenarios start off with the same pertussis incidence levels. This is a check that the simulations run correctly. The minor differences between blunting scenarios in pertussis incidence before the start of the maternal immunization program reflect demographic stochasticity. As shown in previous pertussis models, and in other work, stochastic models fit real-world pertussis incidence better than deterministic models (which do not show any variation between simulations, Domenech de Cellès et al. 2018). To avoid confusion, we clarify this point in the legend of Fig. 4: 'Pertussis incidence before

the start of the maternal immunization program is shown as a control, and variation in incidence between simulations are the result of demographic stochasticity.'

Line 309.

Model results indicate an increase of the pertussis cases among adults after maternal program started. Have you seen any data showing this phenomenon? Has the model included the vaccination impact of the mothers?

This is a good point. Fig S3 shows that if there is an increase in pertussis incidence among the 20+, it is negligible (between 0–5 cases per year per 100,000 individuals, Fig. S3). We clarify these results in l. 323: 'These dynamics persisted in children up to 5 years of age, but dissipated by age 10 years, after which pertussis incidence remained low and the effects of both maternal immunization and blunting on pertussis incidence were negligible (Fig. S3).'

Currently, there is no data indicating an increase in pertussis incidence that is mediated by maternal immunization. In the last decades, many countries have reported increases in pertussis incidence in many age classes and especially in adults (e.g., Domenech de Cellès et al. 2016, Dalby et al. 2016). However, disentangling pre-maternal immunization increases from maternal-immunization-mediated increases remains a challenge, and we are not aware of any study that has undertaken such a challenge. In the revision, we grant a paragraph on this detail in the discussion section, l. 581.

A limitation of the model is that we did not include the impact on the mothers. Therefore, maternal immunization might decrease pertussis incidence in adults. However, in high-income countries, the proportion of pregnant women relative to all adults is low (in 2021 the number of live births among all EU member states ranged from 1.1 to 1.8 live births per 1,000 individuals). Hence, an impact on pertussis incidence, if any, will be small. We discuss the impact of maternal immunization on pertussis incidence and the limitation of our model in the discussion section (now l. 644): "Third, for simplicity, we did not model the direct, protective effect of maternal immunization on mothers—only the indirect effect on their newborns. This direct effect is an additional benefit of maternal immunization, but because in high-income countries the proportion of women giving birth is small relative to the total adult population (2021 range among all EU-member states: 1.1 to 1.8 live births per 1,000 individuals), it is expected to have a minor epidemiological impact."

It is not easy to distinguish the effectiveness of the maternal and primary vaccines as those with maternal vaccinated mothers are likely to vaccinate their infants.

We thank the reviewer for making this point, with which we completely agree. In the discussion, we point out this possible source of bias in empirical studies. We also point out that this bias is absent from our simulations, which has consequences for comparing the model and the empirical studies. We discuss these themes in the discussion section, starting from l. 604. In brief, we mention the possibility of bias in empirical studies, its absence in our simulations, and hence that the RR estimates from our model are, therefore, conservative values.

Line 324. The coverage of the maternal immunization would influence the impact of the blunting on infant vaccination.

This is correct. Therefore, we performed sensitivity analyses to investigate the impact of maternal immunization coverage on the robustness of our results (sensitivity analysis i). Our conclusions remained robust, with maternal immunization coverage ranging from 50% to 90% (Fig. S5 & Fig. S6).

We describe the conclusions of the sensitivity analyses in the last paragraph of the results section, more specifically in l. 386: ‘In our simulations, variation in these parameters sometimes had a substantial impact on the pertussis incidence, e.g., Figs. S7, S9, S11, and S13. However, all the aforementioned conclusions were robust against variation in these parameter values, and the results presented in Figs. S5 to S34 were consistent with those presented in Fig. 5.’

In Fig S6 B&E, I would expect there is no RR change with blunting = zero.

We apologize if the phrasing of this section was confusing. All our simulations found a change in RR over time, irrespective of the levels of blunting. This change occurs as a result of transient dynamics following the roll-out of maternal immunization, consistent with the more general honeymoon dynamics that follow the roll-out of many immunization programs (McLean & Anderson 1988, Keeling & Rohani 2008, p. 295).

Fig. 4A, 4B and Fig. 5 show the same result: transient dynamics occur with or without blunting. We clarify this in the results section l. 360: “Irrespective of the simulated blunting level, the second age class also showed a clear transient phase, lasting at least 5 years (Fig. 4B). During this transient phase, pertussis incidence was first predominantly driven by infants from unvaccinated mothers (Fig. 4C), followed later on by incidence in infants from vaccinated mothers, who dominated once incidence had reached equilibrium (Fig. 4D).”

We also clarify this in the results section l. 388: “During this transient phase, in scenarios with or without blunting, the RR was initially far below 1 and gradually increased after the start of maternal immunization.”

We also clarify this point in the discussion, l. 566: “In the transient phase, however, our model predicts that the baseline value under a no-blunting scenario starts off close to 0 and gradually increases to an equilibrium just below 1.”

Line 326. I thought the average duration of the maternal protection was fixed at 8.7 months.

The results in Figs. 4 and 5 are with a duration of maternal protection of 8.7 months. To check for the robustness of our conclusions, we performed sensitivity analyses, varying the range of the average duration of maternal protection from 4 months to 1 year. Our conclusions remained robust across this range of average duration of maternal protection (Fig. S9, Fig. S10).

To avoid confusion regarding parameter values, we clarified the methods section, which we separated in two sections, one on the parametrization of the base model (l. 775) and one on sensitivity analyses (l. 820).

0 year old mixing is very important. need to mention here about that. What is the take for having similar figures with 7 different contact surveys here?

We now explain that some countries have (i) a larger number of daily contacts in the age groups [5–10), [10–15) and [15–20), and (ii) a larger mixing of [0–5) with adolescents and adults. We also explain that ideally, we would want contact information specifically for <1 year old with other age classes, but that such data are not available from surveys.

We clarify these points in the supplementary information, page 17: ‘Note that (i) some countries have a larger number of daily contacts between the age groups [5,10), [10,15) and [15,20) relative to other age groups, and (ii) some countries have a larger mixing of [0,5) with adolescents or adults (i.e., light red squares parallel to the same-age diagonal squares). Separate contact information for the under 1 year old was not available from the surveys in [12] and [13].’

Reviewer #3 (Remarks to the Author):

This study addressed the impact of maternal pertussis immunization on the risk of diseases in infants using 2 methods: a literature review and a modelling component. While this topic is of interest, there are significant methodological concerns.

Thank you for finding our study interesting. In the rebuttal below, we address your methodological concerns. In brief, we performed a systematic review of the literature and a meta-analysis.

Abstract

Both the short and long abstract do not provide sufficient details on the methods (both the modelling part and review of epidemiological evidence) that were employed in the study, which immune markers were assessed, against endpoints, which study population, and do not provide specific results such as estimated vaccine effectiveness.

Following the manuscript instructions from Nature Communications, we deleted the long abstract.

Following the manuscript instructions from Nature Communications, the short abstract should contain 200 words at the most and should serve as a general introduction to the topic with a brief, non-technical summary of the main results and their implications.

Hence, in the revised abstract, we have attempted to combine both, the journal’s instructions and the reviewer’s comments. The revised abstract is in the manuscript with track changes (l. 15).

In brief, we address your concerns as follows:

(i) Details on the methods: see title clarifying, ‘a systematic review and modeling study’
(ii) Immune markers: because both the systematic review and modeling components are at the epidemiological level, there are no immune markers involved. We clarify this in the abstract as: ‘...we systematically reviewed the literature on the relative risk (RR) of

pertussis after primary immunization of infants born to vaccinated vs. unvaccinated mothers.’

(iii) Endpoints: both the empirical studies from the systematic review and the model quantify the risk of pertussis in vaccinated infants with versus without maternal immunization, which we clarify in the abstract.

(iv) Study population: We expanded the modelling part such that it applies to a range of populations, including high, middle, and low-income countries.

(v) Specific results such as estimated vaccine effectiveness: In the abstract, we now add the estimated vaccine effectiveness from the systematic review, namely (mean: 0.71, 95% CI: 0.38–1.32).

Introduction

The authors use the word ‘demographic’ to describe ‘age group, for example in line 66 ‘Newborns are the demographic most vulnerable to pertussis’. I suggest using the term ‘age group’ instead of demographic.

We have replaced the word ‘demographic’ with ‘age group’ throughout the manuscript.

Much of the significant evidence on pertussis maternal immunization was generated initially in the UK, but very few studies were cited. Hallmark trials should be cited in the introduction. Below a list of relevant articles on this topic.

Thank you. We very much appreciate the extensive work on maternal immunization done by the UK team(s). We have added most of the references on the list below and one other that was not on the list. Note, though, that we had cited several UK studies before and that we show the results of their epidemiological studies in Fig. 1 and Fig. 5.

Some epidemiological terms are incorrectly used. For example, in lines 77-78 the authors stated that ‘estimates of reductions in pertussis incidence ranging from 70 to 95% [10–14]’, while these are reductions in the risk of the disease, rather than reductions in the incidence.

In l. 179, we have replaced ‘incidence’ with ‘reductions in the risk of pertussis infection’.

Lines 80-86: the authors should provide more specific details on the impact of maternal pertussis immunization on pertussis immune response in infants from vaccinated mothers.

We have done so, also following the instructions of reviewer 1, l. 185: “Indeed, several studies and meta-analyses have shown that, after infants received their primary immunization, the antibody concentrations against several pertussis antigens were reduced by 30% to 60% in infants from vaccinated mothers relative to infants from unvaccinated mothers²¹⁻²⁴.”

The rationale of the study is not defined well. A clearer rephrasing of the study objectives is needed.

We have rewritten the last paragraph of the introduction (l. 203), and rephrased the aim of the study in l. 206. “Here, we assess the epidemiological evidence for blunting and estimate the long-term consequences of maternal immunization in three steps. First...”

Lines 99-103 – these are methods/results.

We have replaced this sentence with (l. 207): “First, we perform a systematic review of the literature for estimates of the impact of maternal immunization on the relative risk of pertussis after primary pertussis immunization in infants from vaccinated mothers relative to unvaccinated mothers.”

Lines 106-108 start with ‘Our study shows... and end with reference ‘[24]’. It is not clear, are these results of the current study? Why reference 24 is cited, if these are results of the current analysis.

Following your comment below, we have deleted this sentence in the revised manuscript.

Lines 106-117 these are results/discussion/conclusions and recommendations, that should be placed in the relevant sections rather than in the introduction (which should set the rational/background of the study).

In the revised manuscript, we have deleted the summary of the results at the end of the introduction.

Methods

The authors conducted a narrative review rather than a systematic review. Narrative reviews are subject to biases and this is a major limitation of this manuscript. Systematic reviews on the other hand are done in a structured manner and less prone to bias, while providing critical methodical details, which allow replication of the study.

In the revision, we have performed a systematic review of the literature. In brief, we obtained the same four studies, but the new search is an improvement, also because the literature search is more up-to-date, with one year more of systematic review of the literature until August 25, 2023.

In the methods section, we explained the methods of the systematic review (l. 672), with a section entitled: ‘*Systematic review and meta-analysis of empirical studies on the relative risk of pertussis after maternal and infant primary immunization*’. Due to the length of that section, we only give the title here, and, for the actual text, we refer to the manuscript with track changes.

In the results section, we describe the results of the systematic review, (l. 220), with a section, entitled: ‘*A systematic review of empirical studies on the relative risk of pertussis after maternal and infant primary immunization*’. Due to the length of that section, we only give the title here, and, for the actual text, we refer to the manuscript with track changes.

In the supplementary information, we provide an Excel sheet containing a list of the 69 articles, providing for each study whether it was included or excluded from the results

with the reason for exclusion. These reasons are also described in Fig. 1, the PRISMA figure of the literature search.

It is not clear why the authors conducted a review of studies that assessed the risk of pertussis in infants in relation to maternal immunization without a meta-analysis, which would have much more impact than a review.

Following the systematic review, we ended up with 4 articles that contained information on the relative risk of laboratory-confirmed pertussis in infants from vaccinated versus unvaccinated mothers together with information on the dose of infant primary immunization.

We extracted all the possible information from the four articles and we performed a meta-analysis. Unfortunately, four studies are too few to perform a complete meta-analysis, and we are limited with respect to the predictor variables that we can reliably test, a limitation that we acknowledge in the methods section (l. 725). Hence, we limited the meta-analysis to the variable that is of most interest for the study, namely providing per dose of infant immunization a weighted mean RR of pertussis. We did this following the meta-analysis guidelines provided by Vliechtbauer 2010 and by Shinichi Nakagawa (Nakagawa & Santos 2012), and as we performed in a previous study (Kärkkäinen et al. 2022).

In the revised manuscript, we describe the methods used to perform the meta-analysis in the methods section, l. 693, in the section dedicated to the systematic review. We describe the results of the meta-analysis in the results section, l. 301, in the section dedicated to the systematic review. Due to the length of these sections, we refer to the manuscript with track changes. In the revised manuscript, we have added to Fig. 2 the weighted mean RR in black color.

Fig. 2 Overview of the results from empirical epidemiologic studies of the relative risk of contracting pertussis after 1–3 vaccine doses for primary immunization in infants from vaccinated mothers relative to those of unvaccinated mothers during infant primary immunization. The literature search protocol is shown in Fig. 1, and the data are available in Table S1. The two lines for the UK 3-year study represent two estimates of maternal immunization coverage. References: Australia 2 yrs¹⁶, California 6 yrs¹⁴, UK 3 yrs¹³, and UK 6 yrs¹⁷.

In their review the authors refer to heterogeneity between studies, but not formal test was cited in the methods section.

The term heterogeneity refers to the large confidence intervals around the empirical estimates. To avoid confusion, in the revised manuscript, we have deleted any reference to heterogeneity and we make the point that there is large uncertainty around the estimates, for example:

L. 22: The four studies identified had ≤ 6 years of follow-up and large statistical uncertainty

L. 472: After receipt of the second or third vaccine dose, however, the estimates had large uncertainty, consistent with a range of assumptions about blunting levels.

L. 478: even though the large uncertainty in empirical estimates prevented a definitive conclusion.

L. 509: One of the main takeaways from our review was the large uncertainty around the available empirical estimates

What were the keywords that were used in the literature search?

We explain the keywords used in the literature search in the methods l. 675: 'We searched the literature on Web of Science, PubMed, and Scopus on August 25, 2023. We used the search terms "maternal immunization AND pertussis AND effectiveness OR maternal vaccination AND pertussis AND effectiveness". In Web of Science, we entered these terms under 'Topic', in PubMed we entered these terms under 'Title/Abstract', and in Scopus under 'Article type, Abstract, Keywords'.'

The authors used Google scholar, but what about the Pubmed and Scopus? Why not including these databases in their literature search?

Please see the following reference: Page M J, McKenzie J E, Bossuyt P M, Boutron I, Hoffmann T C, Mulrow C D et al. The PRISMA 2020 statement: an updated guideline for reporting systematic reviews BMJ 2021; 372 :n71 doi:10.1136/bmj.n71.

In our literature search, we now use the three main databases for literature search, Web of Science, PubMed, and Scopus. We explain the databases in the Methods section l. 675: 'We searched the literature on Web of Science, PubMed, and Scopus on August 25, 2023.'

We have revised Page's PRISMA figure (Fig. 1), in which we show these three databases and we follow the guidelines provided by Page et al. 2021.

Fig. 1 Prisma flow chart of the literature search for empirical estimates of the relative risk (RR) of contracting pertussis following maternal immunization in children who had received at least one dose of infant primary immunization.

Model parametrization: I am concerned that the model parametrization was based on one study (reference# 3) using data from Massachusetts. what is the validity of the approach and potential generalizability of the findings.

We understand your concerns and we have carried out a number of sensitivity analyses to address this limitation. These include among others (i) changes in maternal immunization coverage (base value: 70%, sensitivity analyses: 50%–90%), (ii) changes in primary immunization coverage (base value: 90%, sensitivity analyses: 70% and 80%), (iii) changes in the infant primary immunization program (base value: 3 months, sensitivity analyses: 2 months until 9 months), and (iv) different contact matrices from 8 European countries.

In addition to the above, one way to account for differences between populations is to change the model's transmission parameters (q_1), with low versus high transmission reflecting populations with low and high pertussis circulation respectively. We simulated a $\pm 50\%$ variation in the basic reproduction number, to model different transmission levels in a range of populations.

In the methods section, we clarify these points with the section "Sensitivity analyses" (l. 818).

We discuss the sensitivity analysis at length in the results section (l. 424), with a subsection entitled: '*Sensitivity analyses*'. Due to the length of that section, we only give the title here, and, for the actual text, we refer to the manuscript with track changes.

In the supplementary information, we show the figures of the sensitivity analyses. These are too long to add in the rebuttal, but in brief, we show the results of the sensitivity

analyses in Figs. S5 to S34. Again, variations in these parameter values create populations that deviate in many ways from that in Massachusetts and our take-home messages are robust against changes in all these parameter values

There are other approaches, ranked at a higher level of evidence (e.g., cohort studies, case-control studies) for estimating vaccine effectiveness than the ‘screening method’. The rationale for using this approach is not clear.

Our main motivation to use this method was because two out of four empirical studies have used this method (Amirthalingam et al. 2016, Amirthalingam et al. 2022), and hence the method used in our simulations best matches that used in empirical studies.

We are aware of the limitations of this method in real-world settings (ref. 67), but in simulations with perfect and complete information, these do not apply. In our simulations, we have validated the screening method because the effectiveness that comes out from the simulations is the same as the input value of the simulation (Fig. S2).

In the methods section, we clarify our choice in l. 864: “To connect the model outputs with the empirical estimates of relative risk (RR)^{13,14,16,17}, we calculated the risk of contracting pertussis in infants born to vaccinated mothers relative to that of infants born to unvaccinated mothers using the same approach as in^{13,17}. These studies used the screening method following⁶⁶,”

In the methods section, we also clarify the validation of the method in l. 877: “We validated the RR estimates of the simulations by setting the half-life of maternally derived protection to infinity, which resulted in a RR of 0.05 in newborns and matched the initial 95% effectiveness of maternal immunization against pertussis in unvaccinated newborns fixed in the simulations (Fig. S2, Table S3). Hence, while the screening method may have several limitations in real-world settings⁶⁷, it provided reliable estimates of vaccine effectiveness in the context of our model.”

The model assumptions of 95% coverage of the primary series and booster vaccination might be too optimistic.

We show the main results with coverage for the primary series at 90%. In addition, we have performed a sensitivity analysis with the coverage of the primary series at 70% and 80%. The range of 70% to 90% reflects the primary immunization coverage in many high-income countries with maternal immunization. We discuss the results of this sensitivity analysis in the last paragraph of the results section. We show the figures in the Supplementary information (Fig S7 & S8, see below in the rebuttal). In brief, the conclusions of our study are robust within the window of 70%–90% primary immunization coverage.

We have added this sensitivity analysis in the methods section (l. 822): “(i) infant immunization coverage, changed from a baseline value of 90% to 70% and 80% (Fig. S7, S8);”

Fig S7: Sensitivity analysis 2: Impact of changing the primary immunization coverage (base value: 90%, tested values: 70% and 80%). The Y-axis shows the pertussis incidence. Note the different scales of the Y-axes between panels. 90% maternal immunization coverage is the value shown in Fig. 4. Everything else is equal as in Fig. 4.

Fig S8: Sensitivity analysis 2: Impact of changing the primary immunization coverage (base value: 90%, tested values: 70% and 80%). The Y-axis shows the RR of pertussis in infants from vaccinated mothers relative to unvaccinated mothers at (A, B, C) 70% primary immunization coverage and at (D, E, F) 80% primary immunization coverage. 90% primary immunization coverage is the value shown in Fig. 5. Everything else is equal as in Fig. 5.

Relevant references:

1: Grassly NC, Andrews N, Cooper G, Stephens L, Waight P, Jones CE, Heath PT, Calvert A, Southern J, Martin J, Miller E. Effect of maternal immunisation with multivalent vaccines containing inactivated poliovirus vaccine (IPV) on infant IPV immune response: A phase 4, multi-centre randomised trial. *Vaccine*. 2023 Feb 10;41(7):1299-1302. doi: 10.1016/j.vaccine.2023.01.035.

We refer to this paper in our manuscript.

2: Sapuan S, Andrews N, Hallis B, Hole L, Jones CE, Matheson M, Miller E, Snape MD, Heath PT. An observational, cohort, multi-centre, open label phase IV extension study comparing preschool DTAP-IPV booster vaccine responses in children whose mothers were randomised to one of two pertussis-containing vaccines or received no pertussis-containing vaccine in pregnancy in England. *Vaccine*. 2022 Nov 22;40(49):7050-7056. doi: 10.1016/j.vaccine.2022.10.005.

We refer to this paper in our manuscript.

3: Oguti B, Ali A, Andrews N, Barug D, Anh Dang D, Halperin SA, Thu Hoang HT, Holder B, Kampmann B, Kazi AM, Langley JM, Leuridan E, Madavan N, Maertens K, Maldonado H, Miller E, Munoz-Rivas FM, Omer SB, Pollard AJ, Rice TF, Rots N, Sundaram ME, Wanlapakorn N, Voysey M. The half-life of maternal transplacental antibodies against diphtheria, tetanus, and pertussis in infants: an individual participant data meta-analysis. *Vaccine*. 2022;40(3):450-458. doi: 10.1016/j.vaccine.2021.12.007.

We refer to this paper several times in our manuscript.

4: Jones CE, Calvert A, Southern J, Matheson M, Andrews N, Khalil A, Cuthbertson H, Hallis B, England A, Heath PT, Miller E. A phase IV, multi-centre, randomized clinical trial comparing two pertussis-containing vaccines in pregnant women in England and vaccine responses in their infants. *BMC Med*. 2021;19(1):138. doi: 10.1186/s12916-021-02005-5

We refer to this paper in our manuscript.

5: Amirthalingam G, Campbell H, Ribeiro S, Fry NK, Ramsay M, Miller E, Andrews N. Sustained Effectiveness of the Maternal Pertussis Immunization Program in England 3 Years Following Introduction. *Clin Infect Dis*. 2016 Dec 1;63(suppl4):S236-S243. doi: 10.1093/cid/ciw559.

We refer to this paper many times in our manuscript, including in Fig. 2 and Fig 5.

6: Choi YH, Campbell H, Amirthalingam G, van Hoek AJ, Miller E. Investigating the pertussis resurgence in England and Wales, and options for future control. *BMC Med*. 2016 Sep 1;14(1):121. doi: 10.1186/s12916-016-0665-8.

This is not a study on maternal immunization, but comparing whole-cell and acellular primary immunization, and therefore beyond the scope of our study.

7: Kent A, Ladhani SN, Andrews NJ, Matheson M, England A, Miller E, Heath PT; PUNS study group. Pertussis Antibody Concentrations in Infants Born Prematurely to Mothers Vaccinated in Pregnancy. *Pediatrics*. 2016 Jul;138(1):e20153854. doi: 10.1542/peds.2015-3854

This paper is on premature births, which is beyond the scope of our study.

8: van Hoek AJ, Campbell H, Amirthalingam G, Andrews N, Miller E. Cost-effectiveness and programmatic benefits of maternal vaccination against pertussis in England. *J Infect*. 2016 Jul;73(1):28-37. doi:10.1016/j.jinf.2016.04.012.

This is a valuable cost-effectiveness study, but cost-effectiveness is beyond the scope of our study.

9: Ladhani SN, Andrews NJ, Southern J, Jones CE, Amirthalingam G, Waight PA, England A, Matheson M, Bai X, Findlow H, Burbidge P, Thalasselis V, Hallis B, Goldblatt D, Borrow R,

Heath PT, Miller E. Antibody responses after primary immunization in infants born to women receiving a pertussis-containing vaccine during pregnancy: single arm observational study with a historical comparator. *Clin Infect Dis.* 2015 Dec 1;61(11):1637-44. doi: 10.1093/cid/civ695.

This paper is used in the meta-analysis we refer to in our manuscript.

10: Amirthalingam G, Andrews N, Campbell H, Ribeiro S, Kara E, Donegan K, Fry NK, Miller E, Ramsay M. Effectiveness of maternal pertussis vaccination in England: an observational study. *Lancet.* 2014 Oct 25;384(9953):1521-8. doi:10.1016/S0140-6736(14)60686-3.

We refer to this paper in our manuscript.

In addition, we have also added another UK reference, l. 897: “Amirthalingam, G., Gupta, S. & Campbell, H. Pertussis immunisation and control in England and Wales, 1957 to 2012: a historical review. *Euro Surveill.* **18**, ii=20587 (2013).”

References for the rebuttal (in alphabetical order following the first author’s last name):

Ahmed R, Oldstone MB, Palese P. (2007) Protective immunity and susceptibility to infectious diseases: lessons from the 1918 influenza pandemic. *Nat Immunol*, 8:1188–93.

Amirthalingam G, Campbell H, Ribeiro S, Stowe J, Tessier E, Litt D, et al. 2023. Optimization of timing of maternal pertussis immunization from 6 years of postimplementation surveillance data in England. *Clinical Infectious Diseases* 76(3):e1129–e1139.

Brett TS & Rohani P. Transmission dynamics reveal the impracticality of COVID-19 herd immunity strategies. *Proc. Natl. Acad. Sci. U.S.A.* 117, 25897–25903.

Causey, K et al. 2021 Estimating global and regional disruptions to routine childhood vaccine coverage during the COVID-19 pandemic in 2020: a modelling study. *The Lancet*, 398, 10299: 522 – 534.

Dabrera G, Amirthalingam G, Andrews N, Campbell H, Ribeiro E Sonia Kara, Fry NK, et al. 2015. A case-control study to estimate the effectiveness of maternal pertussis vaccination in protecting newborn infants in England and Wales, 2012–2013. *Clinical Infectious Diseases* 60(3):333–337.

Domenech de Cellès, M., Magpantay, F. M. G., King, A. A. & Rohani, P. 2016. The pertussis enigma: reconciling epidemiology, immunology and evolution. *Proc. Biol. Sci. B* 283, 20152309.

Domenech de Cellès M, Magpantay FMG, King AA, Rohani P. 2018. The impact of past vaccination coverage and immunity on pertussis resurgence. *Sci Transl Med.* 10(434):eaaj1748.

Domenech de Cellès, M., Rohani, P. & King, A. A. 2019. Duration of Immunity and Effectiveness of Diphtheria-Tetanus-Acellular Pertussis Vaccines in Children. *JAMA Pediatr.* 173, 588–594.

Funk S, Willem L. socialmixr: Social Mixing Matrices for Infectious Disease Modelling; 2022. Available from: <https://CRAN.R-project.org/package=socialmixr>.

Kandeil W, van den Ende C, Bunge EM, Jenkins V, Ceregido MA, Guignard A. 2020. A systematic review of the burden of pertussis disease in infants and the effectiveness of maternal immunization against pertussis, *Expert Review of Vaccines*, 19(7): 621-638.

Kärkkäinen T, Briga M, Laaksonen T, Stier A. 2022. Within-individual repeatability in telomere length: a meta-analysis in non-mammalian vertebrates. *Molecular Ecology*, 31, 6339-6359.

Keeling, M.J. and Rohani, P. 2008. *Modeling Infectious Diseases in Humans and Animals*, Princeton University Press, Princeton.

Kim S, Headley TY, Tozan Y. 2022. Universal healthcare coverage and health service delivery before and during the COVID-19 pandemic: A difference-in-difference study of childhood immunization coverage from 195 countries. *PLoS Med* 19(8): e1004060.

Lai X, Zhang H, Pouwels KB, Patenaude B, Jit M, Fang H. 2023. Estimating global and regional between-country inequality in routine childhood vaccine coverage in 195 countries and territories from 2019 to 2021: a longitudinal study. *EClinicalMedicine*, 60: 102042.

Mantzari E, Rubin GJ, Marteau TM. 2020. Is risk compensation threatening public health in the covid-19 pandemic? *BMJ* 370:m2913.

McLean, R. Anderson, (1988) Measles in developing countries. Part ii. The predicted impact of mass vaccination, *Epidemiol. Infect.*, 100, 419–442.

Peltzman, S. 1975. The effects of automobile safety regulation. *J. Polit. Econ.* 83(4), 677–725.

Metcalf CJE, Wesolowski A, Winter AK, Lessler J, Cauchemez S, Moss WJ, et al. (2020) Using Serology to Anticipate Measles Post-honeymoon Period Outbreaks. *Trends in Microbiology* 28 (8), 597-600.

Mossong J, Hens N, Jit M, Beutels P, Auranen K, et al. (2008) Social Contacts and Mixing Patterns Relevant to the Spread of Infectious Diseases. *PLOS Medicine* 5(3): e74.

Page M J, McKenzie J E, Bossuyt P M, Boutron I, Hoffmann T C, Mulrow C D et al. (2021) The PRISMA 2020 statement: an updated guideline for reporting systematic reviews *BMJ* 372 :n71.

Prem K, Cook AR, Jit M (2017) Projecting social contact matrices in 152 countries using contact surveys and demographic data. *PLOS Computational Biology* 13(9): e1005697.

Nakagawa, S., Santos, E.S.A. 2012. Methodological issues and advances in biological meta-analysis. *Evol Ecol* 26, 1253–1274.

Perrett KP, Halperin SA, Nolan T, Pancorbo CM, Tapiero B, et al. 2020. Immunogenicity, transplacental transfer of pertussis antibodies and safety following pertussis immunization during pregnancy: Evidence from a randomized, placebo-controlled trial. *Vaccine* 38 (8): 2095-2104.

Rane MS, Rohani P, Halloran ME. Association of Diphtheria-Tetanus–Acellular Pertussis Vaccine Timeliness and Number of Doses With Age-Specific Pertussis Risk in Infants and Young Children. *JAMA Netw Open*.2021;4(8):e2119118.

Riise, Ø.R., Laake, I., Bergsaker, M.A.R. et al. (2015). Monitoring of timely and delayed vaccinations: a nation-wide registry-based study of Norwegian children aged < 2 years. *BMC Pediatr* 15, 180.

Riolo MA, King AA, Rohani P. Can vaccine legacy explain the British pertussis resurgence? *Vaccine*. 2013 Dec 2;31(49):5903-8.

Skoff TH, Blain AE, Watt J, Scherzinger K, McMahon M, Zansky SM, et al. 2017. Impact of the US maternal Tetanus, Diphtheria, and Acellular Pertussis vaccination program on preventing Pertussis in infants <2 months of age: A case-control evaluation. *Clinical Infectious Diseases* 65(12):1977–1983.

Viechtbauer, W. 2010. Conducting Meta-Analyses in R with the metafor Package. *Journal of Statistical Software*, 36(3), 1–48.

REVIEWER COMMENTS

Reviewer #1 (Remarks to the Author):

Thank you for the opportunity to review the revised version and the response letter.

The reviewers have adequately responded to my comments and clarifications.

I have no main concerns.

I have 2 more minor comments:

1) Lines 370-373: this sentence is not accurate. There are no vaccines trialed in pregnant women for TB, mumps and rotavirus. Please restrict this to pertussis and streptococcus pneumoniae where we have evidence

This sentence was referring to blunting rather than vaccine development. To avoid confusion, we have rephrased this sentence as (l. 495): "In addition to the Tdap and RSV vaccines, maternal immunization is recommended or under development for influenza, COVID-19, polio and several other infections^{31–33}"

Reviewer's response: this is not entirely true, I would remove polio virus. For COVID-19, it is already implemented.

2) 381-384: it is important to report the meta-analysis showing that the higher antibody levels at primary immunization the lower the immune responses suggesting delaying primary vaccination (see PMID: 34598822), and also that other factors did not affect infants' immune responses

We have added the reference here (l. 513): "Given the fast waning of maternal antibodies and the negative association between the maternal antibody titers and the blunting of infant immune responses³⁸, delaying infant primary immunization by a few months might greatly reduce the maternal blunting of infant immunization³⁶."

Reviewer's response: adding "negative" is confusing and the message is conveyed better by removing "negative" so it reads :

"Given the fast waning of maternal antibodies and the association between the maternal antibody titers and the blunting of infant immune responses³⁸....."

Reviewer #2 (Remarks to the Author):

I would like to thank the authors to answer to all of my questions. I have one comment and two more questions.

Comment

Regarding the contact survey among infants, I would like to recommend a contact survey study among infants done in England by van Hoek et al. (2013) as you used the GB POLYMOD:

van Hoek AJ, Andrews N, Campbell H, Amirthalingam G, Edmunds WJ, Miller E. The social life of infants in the context of infectious disease transmission; social contacts and mixing patterns of the very young. *PLoS One* 2013; 8(10): e76180.

Questions

Re the degree of the vaccine protection: You ignored the possibility of failure in degree (or leakiness) among individuals protected by both maternal and primary vaccinations. The key finding from the study by Warfel et al. (2014), indicated strong evidence of a lower degree of acellular vaccine protection and suggesting the transmissibility of acellular vaccinated baboons to susceptible baboons.

Warfel JM, Merkel TJ. The baboon model of pertussis: effective use and lessons for pertussis vaccines. *Expert Rev Vaccines*. 2014;13:1241–52.

Duration of the maternal and primary vaccine protection: According to the equations and Fig S1 in Supplementary Appendix, the duration of maternal vaccine protection ($1/\tau$) for M is merged to the duration of the primary vaccine protection ($1/\alpha_V$) when a proportion of M received the primary vaccine (V_2^M). Conversely, those who did not receive the primary vaccine would lose the maternal vaccine protection with τ in M^V . Therefore, the blunting impact is solely dependent on the duration of maternal protection in M^V . Is that right?

Reviewer #1

Thank you for the opportunity to review the revised version and the response letter. The reviewers have adequately responded to my comments and clarifications.

I have no main concerns.

I have 2 more minor comments:

1) Lines 370-373: this sentence is not accurate. There are no vaccines trialed in pregnant women for TB, mumps and rotavirus. Please restrict this to pertussis and streptococcus pneumoniae where we have evidence

This sentence was referring to blunting rather than vaccine development. To avoid confusion, we have rephrased this sentence as (l. 495): “In addition to the Tdap and RSV vaccines, maternal immunization is recommended or under development for influenza, COVID-19, polio and several other infections³¹⁻³³”

Reviewer's response: this is not entirely true, I would remove polio virus. For COVID-19, it is already implemented.

We have removed polio and restricted this sentence to those vaccines that are already implemented or recommended, l. 258: “In addition to the Tdap and RSV vaccines, maternal immunization is implemented or recommended for influenza, COVID-19, and several other infections³¹⁻³³”.

2) 381-384: it is important to report the meta-analysis showing that the higher antibody levels at primary immunization the lower the immune responses suggesting delaying primary vaccination (see PMID: 34598822), and also that other factors did not affect infants' immune responses

We have added the reference here (l. 513): “Given the fast waning of maternal antibodies and the negative association between the maternal antibody titers and the blunting of infant immune responses³⁸, delaying infant primary immunization by a few months might greatly reduce the maternal blunting of infant immunization³⁶.”

Reviewer's response: adding “negative” is confusing and the message is conveyed better by removing “negative” so it reads :

“Given the fast waning of maternal antibodies and the association between the maternal antibody titers and the blunting of infant immune responses³⁸”

We agree. We have removed the “negative” and rephrased the sentence as suggested by the reviewer (l. 270).

Reviewer #2

Reviewer #2 (Remarks to the Author):

I would like to thank the authors to answer to all of my questions. I have one comment and two more questions.

Comment

Regarding the contact survey among infants, I would like to recommend a contact survey study among infants done in England by van Hoek et al. (2013) as you used the GB POLYMOD: van Hoek AJ, Andrews N, Campbell H, Amirthalingam G, Edmunds WJ, Miller E. The social life of infants in the context of infectious disease transmission; social contacts and mixing patterns of the very young. PLoS One 2013; 8(10): e76180.

Thank you for mentioning this important study, which we now cite in the manuscript l. 529.

Questions

Re the degree of the vaccine protection: You ignored the possibility of failure in degree (or leakiness) among individuals protected by both maternal and primary vaccinations. The key finding from the study by Warfel et al. (2014), indicated strong evidence of a lower degree of acellular vaccine protection and suggesting the transmissibility of acellular vaccinated baboons to susceptible baboons.

Warfel JM, Merkel TJ. The baboon model of pertussis: effective use and lessons for pertussis vaccines. Expert Rev Vaccines. 2014;13:1241–52.

We thank the reviewer for raising this important question. We are well aware of these experimental results, and we pointed out elsewhere—starting with our reply to the original PNAS paper (1)—that they were inconsistent with the large body of epidemiological evidence showing that acellular pertussis (aP) vaccines markedly reduce infection risk and induce strong herd immunity (see our review listing this evidence (2)). As we explained in the main text, our work is based on a previous study during 1990–2005 in Massachusetts, where we used likelihood-based inference to test a range of hypotheses about pertussis vaccines (3). In particular, we explicitly tested the hypothesis of vaccine leakiness, finding no evidence for it when fitting to the whole period (Table 1) or only to the post-aP period (Table S9). In a later study, we also showed that aP vaccines conferring initially high, but subsequently waning protection against infection could explain more recent epidemiological observations in the US (4). Hence, although we recognize the persistence of disagreements in the field, our model is realistic, and its representation of aP properties is grounded in evidence.

In the manuscript, we clarified the epidemiological basis regarding leaky vaccines in l. 447: “Previous results in⁴ found no evidence for failure in degree (or leakiness, i.e., when vaccine-induced protection is imperfect and vaccinees remain susceptible to infection, but at a lower degree than unvaccinated individuals), and hence we ignored this possibility.”

Duration of the maternal and primary vaccine protection: According to the equations and Fig S1 in Supplementary Appendix, the duration of maternal vaccine protection ($1/\tau$) for M is merged to the duration of the primary vaccine protection ($1/\alpha_V$) when a proportion of M received the primary vaccine ($V_2^{(M)}$). Conversely, those who did not receive the primary vaccine would lose the maternal vaccine protection with τ in $M^{(V)}$. Therefore, the blunting impact is solely dependent on the duration of maternal protection in $M^{(V)}$. Is that right?

The reviewer is correct, yes. Specifically, the effect of blunting (parameter b_1) is captured in the modified vaccine effectiveness parameter $\bar{\epsilon} = \epsilon(1 - b_1)$, where ϵ is the vaccine effectiveness in the absence of blunting. As newborns protected by maternal antibodies (compartments M_1) age and get vaccinated (at coverage $p_1(t)$), the vaccine either takes (transitions $M_1 \rightarrow V_2^{(M)}$) or doesn't (transitions $M_1 \rightarrow M_2^{(V)}$). In the latter case, even though the vaccine didn't take, the residual protection from maternal antibodies remains, subsequently waning at the rate τ .

References

1. Domenech de Cellès M, Riolo MA, Magpantay FMG, Rohani P, King AA. Epidemiological evidence for herd immunity induced by acellular pertussis vaccines. *Proc Natl Acad Sci U S A*. 2014 Feb 18;111(7):E716–7.
2. Domenech de Cellès M, Magpantay FMG, King AA, Rohani P. The pertussis enigma: reconciling epidemiology, immunology and evolution. *Proc Biol Sci* [Internet]. 2016 Jan 13;283(1822). Available from: <http://dx.doi.org/10.1098/rspb.2015.2309>
3. Domenech de Cellès M, Magpantay FMG, King AA, Rohani P. The impact of past vaccination coverage and immunity on pertussis resurgence. *Sci Transl Med* [Internet]. 2018 Mar 28;10(434). Available from: <http://dx.doi.org/10.1126/scitranslmed.aaj1748>
4. Domenech de Cellès M, Rohani P, King AA. Duration of Immunity and Effectiveness of Diphtheria-Tetanus-Acellular Pertussis Vaccines in Children. *JAMA Pediatr*. 2019 Jun 1;173(6):588–94.